# Influences of Lake Malawi

# on the spatial and diurnal variability of local precipitation

**Shunya Koseki[1] and Priscilla A. Mooney[2]**

1: Geophysical Institute, University of Bergen, Bjerknes Centre for Climate Research, Bergen, Norway

2: NORCE Norwegian Research Centre, Bjerknes Centre for Climate Research, Bergen, Norway

Corresponding author: Shunya Koseki

Email: Shunya.Koseki@gfi.uib.no

Address: Allégate 70, 5007, Bergen, Norway

**Abstract**

2        We investigate how the intensity and spatial distribution of precipitation varies

around Lake Malawi on a diurnal time scale, which can be valuable information for
water resource management in tropical southeastern African nations. Using a state-of-
the-art satellite product and regional atmospheric model, the well-defined diurnal
cycle is detected around Lake Malawi with harmonic and principle component
analyses: the precipitation is intense during midnight to morning over Lake Malawi
and the precipitation peaks in the daytime over the surrounding area. This diurnal
cycle in the precipitation around the lake is associated with the lake-land breeze
circulation. Comparisons between the benchmark simulation and an idealized
simulation in which Lake Malawi is removed, reveals that the diurnal variations in
precipitation are substantially amplified by the presence of Lake Malawi. This is most
evident over the lake and surrounding coastal regions. Lake Malawi also enhances the
lake-land breeze circulation; the nocturnal lakeward land breeze generates surface
convergence effectively and precipitation intensifies over the lake. Conversely, the
daytime landward lake breeze generates the intense divergence over the lake and
precipitation is strongly depressed over the lake. The lake-land breeze and the
background vapour enriched by Lake Malawi drives primarily  a diurnal variation in
the surface moisture flux divergence/convergence over the lake and surrounding area
which contributes to the diurnal cycle of precipitation in this region.

**1. Introduction**

A key climatological characteristic of tropical southeastern Africa is the manifestation of dry and wet seasons induced by the meridional march of the Inter-tropical Convergence Zone (ITCZ). This seasonal movement in the ITCZ is associated with the southwesterly Indian summer and northeasterly winter monsoons (e.g., Camberlin 1997; Viste and Sorteberg, 2013; Jury 2016; Diallo et al., 2018; Koseki and Bhatt, 2018) as shown in Figs. S1a-l. In summer (May to September), tropical southeasten Africa is covered entirely with a moisture flux divergence (Fig. S1m) and consequently, a dry season falls on this region. The northeastward moisture flux provides some of the summer precipitation over northeastern Africa and South Asia (e.g., Segele et al., 2009a; Viste and Sorteberg, 2013; Gleixner et al., 2017; Bohlinger et al., 2017). Conversely, the southwestward Indian winter monsoon generates a large convergence of vertically-integrated moisture flux over the tropical southeast of Africa  (November to March, as shown Fig. S1n) bringing a wet season to this region. This monsoon-brought precipitation is very important for the regional economy and society of the southeastern African nations such as Tanzania, Mozambique, Madagascar, and Malawi where their economies depend highly on rain-fed agriculture.

Variability in hourly rainfall is also dominant over Southeastern Africa. It is controlled largely by a diurnal cycle due to the thermal heat contrast between water surface and land surface in the tropics (e.g., Estoque, 1967; Mak and Walsh, 1976; Kikuchi and Wang, 2008; Teo et al., 2011; Koseki et al., 2013; Jury, 2016). The diurnal cycle is observed ubiquitously around the tropical coastal areas since the thermal contrast between coastal land and ocean during daytime and nighttime,

induces the sea and land breeze circulation (e.g., Kitoh and Arakawa, 2005; Kikuchi
and Wang, 2008; Teo et al., 2011; Diro et al., 2012; Koseki et al., 2013). Steep terrain
and land-lake contrast also generate the similar diurnal variations in precipitation.
These variations are associated with the diurnal cycle of mountain-valley and lake-
land breeze systems (e.g., Keen and Lyons, 1978; Joseph et al., 2008; Stivari et al,
2003; Crosman and Horel, 2010; Koseki et al., 2018). Such information on diurnal
variation in precipitation is highly important for efficient water resource management
in nations with economies that depend strongly on rain.
Lake Malawi, located at 12.11°S and 34.22°E (Fig. S1), is the third largest of
the African Great Lakes and ninth in the world having an area of 29,600 km$^2$, a
maximum width of 75km, and a maximum length of 560km. Lake Malawi is an
important water resource for surrounding tropical southeastern African nations such
as Malawi, Mozambique and Tanzania (Kumambala and Ervine, 2010). In particular,
a large part of agriculture and energy in Malawi originates from the water resource of
Lake Malawi and the Shire River which flows from the lake; all of the national
hydropower stations are built on the Shire River (a total installation capacity of
280MW; Kumambala and Ervine, 2010) and the largest national sugar plantations are
supplied with water from the Shire River. Societies along the Shire River and
surrounding Lake Malawi are exposed to high risks of flooding during the rainy
season (November to March, Fig. 1) when the lake level is high due to rainfall over
the lake (e.g., Neuland, 1984; Schäfer et al., 2015). Regarding other aspects, Lake
Malawi is an important fishing resource in Malawi and it has a very unique ecosystem
and biodiversity (e.g., Weyl et al., 2010). Lake Malawi itself plays an important role
in the regional climate system. Diallo et al. (2018) performed climate simulations
with a state-of-the-art regional climate model and suggested that Lake Malawi is a
water source for regional precipitation (over the lake and surrounding area) via
intense latent heat flux release from the lake surface.
Although Diallo et al. (2018) have investigated the role of Lake Malawi for
monthly time scales, little is known about the diurnal cycle of rainfall around Lake
Malawi and the lakes's influence on the diurnal cycle. In general, the African Great
Lakes play an important role for the regional hydrological weather and climate system
as a large water source. For example, Thiery et al. (2016) showed that Lake Victoria
(area of 59,947km$^2$), which is the largest African Great Lake, triggers extreme
thunderstorm over the lake during nighttime. Other exampes include severe
snowstorms around the Great Lakes of North America (area of 244,106km$^2$) (e.g.,
Sousonis and Mann, 2000; NOrato et al., 2013), and local precipitation induced by
Lake Chad (area of 25,000km$^2$) (Lauwaet et al., 2012). Since Lake Malawi, a large
water body (29,600km$^2$), is located in the tropics, the region can be affected by the
strong diurnal cycle of incoming solar radiation (e.g., Crosman and Horel, 2010). This
is the main driver of the diurnal variations in precipitation and local breeze systems.
Although it is expected that Lake Malawi can drive local circulation in response to the
diurnal solar radiation, the lake's role in the diurnal cycle of precipitation is less clear
and poorly understood. This is partly due to the lack of tools to study this topic but
recent developments in the resolution of numerical models now permit such
investigations.
This study aims to investigate the regional diurnal cycle of precipitation in the
rainy season (November to March) and quantify the effects of Lake Malawi on the
diurnal cycle of precipitation using state-of-the-art observational products and
numerical regional model. Using a satellite product with a relatively coarse spatial
resolution, a climatological diurnal cycle is overviewed and a case study of November
to March in 2014/15 is investigated using a higher resolution satellite product for the
purpose of evaluating the numerical simulation.

The rest of this paper is structured as follow: Section 2 gives the details of

observational data and numerical model used in this study and statistical
methodologies to investigate the diurnal variations. Section 3 provides the results of
the statistical analysis on the observations and numerical simulation including an
assessment of the modeled diurnal cycle. Moreover, the results of an idealized
numerical experiment will be used to elucidate the physical mechanisms that underlie
Lake Malawi's role in the diurnal cycle of precipitation around the lake. Section 4
will discuss the details of the simulation results focusing on the quantification of the
influence of Lake Malawi and finally, we will summarize this study in Section 5.

**2. Data, Model, and Methodology**
*2.1 Observational Data*

Satellite observations are obtained from both the Tropical Rainfall Measuring

Mission (Huffman et al., 2007) version 3B42 (TRMM 3B42) and the Global
Precipitation Measurement (GPM, Skofronick-Jackson et al., 2017) mission data
(Level-3). TRMM 3B42 has a high temporal coverage (1998-2014) which facilitates a
climatological overview of the diurnal cycle over Lake Malawi. However, the spatial
resolution of TRMM 3B42 (0.25°) prohibits its use in the analysis of the spatial
characteristics of the diurnal cycle over the lake and its shores. This difficulty is
overcome by using GPM, the successor to TRMM 3B42, which has a higher spatial
resolution of 0.1°. This facilitates a more detailed study of spatial variations in the
diurnal cycle of precipitation. The temporal resolution of the original GPM Level-3
data is every 30 minutes which is averaged to hourly rainfall in this study.

*2.2 Weather Research and Forecasting (WRF) Model*

The Advanced Research Weather Research and Forecasting (hereafter referred to WRF, Skamarock et al., 2007) model version 3.9.1 is used to investigate the diurnal variations around Lake Malawi. The domains used in all simulations are shown in Figure 1a. The outer domain covers southeastern Africa, -20.74902°S to -2.958107°S and 23.3115°E to 44.0885°E with 15km grid spacings (171×117 grids) and the inner domain is centred on Lake Malawi, -15.87943°S to -8.219772°S and 32.22042°E to 37.06839°E with 3km grid spacing (155×250 grids), respectively (Fig. 2a). Both domains have 56 vertical layers. The outer domain is forced laterally with 6 hourly ERA-Interim (Dee et al., 2011) data which has a grid spacing of 0.75° and at the lower boundary by the daily Optimum Interpolated Sea Surface Temperature (OISST, Reynolds et al., 2007) which has a grid spacing of 0.25°. The inner domain is forced laterally by the outer domain of WRF (the outer domain of WRF does not interact with the inner domain).

The following physical schemes are used in our WRF simulations: the WRF Single-moment (WSM) 6-class scheme for microphysics (Hong and Lim, 2006) and the Yonsei University parameterization for the Planetary Boundary Layer (PBL; Hong et al., 2006). The longwave and shortwave radiative forcing are parameterized by the Rapid Radiative Transfer Model (Mlawer et al., 1997) schemes. Betts-Miller-Janjíc (Janjíc, 1994) scheme is used for parameterizing convective processes in the outer domain only; cumulus parameterization is switched off in the convection permitting inner domain. A study of the sensitivity of precipitation in this region to the convective schemes used in the outer domain showed that simulations using the Betts-Miller-Janjíc scheme reproduced the observed precipitation over land better

than simulations using the Kain-Fritsch (Kain, 2004) scheme (not shown). Therefore,
the Betts-Miller-Janjíc scheme is chosen for the outer domain in this study with no
cumulus scheme used in the inner, high-resolution domain. Over the land and lake
grids that are based on MODIS landuse data, the NOAH land surface model
consisting of 4-layers (Chen and Dudhia, 2001a, b) and the 9-layer lake model (Xu et
al., 2016) are implemented and air-land/lake interactions are active in the simulations.

With the model configurations above, a control experiment is initialized on the

1$^{st}$ January 2014 at 00 UTC of ERA-Interim for atmosphere and land surface and
integrated until April 1$^{st}$ of 2015 (referred to WRF-CTL, hereafter). This run will
compliment the observations to gain insights into the diurnal variations around Lake
Malawi. In a second experiment, the grid boxes over Lake Malawi are converted from
water to land grid boxes (Figs. 1b and c). This facilitates an exploration of the role of
Lake Malawi on the local diurnal variations (called WRF-NOLM in the rest of the
paper). Due to this conversion, some land surface properties are modified in WRF-
NOLM: the landuse index of the converted grids is set to be savanna which is the
most dominant landuse category in the inner domain of WRF experiments. The soil
type of the converted grids is also replaced with sandy clay loam which is the
majority soil type for the savanna grids in the inner domain. Additionally, the surface
albedo over Lake Malawi grids is set to a value of albedo averaged over the savanna
grids in the inner domain. Finally, the soil moisture and temperature of the converted
grids are initialized by averaged value of savanna grids. These modifications are done
only in the inner domain to avoid any modulations in larger-scale meteorological and
hydrological quantities associated with the absence of Lake Malawi. All settings of
the outer domain of WRF-NOLM are exactly the same as those of WRF-CTL.
We analyze the hourly output of the 5 months from November in 2014 to
March in 2015; that is the first 10 months are designated as a spin-up period for
initialising the land surface following the methodologies of Cosgrove et al. (2002) and
Chen et al. (2007). In particular, in WRF-NOLM, the soil moisture and temperature
are initialized with an averaged value, which is to a large extent artificial. Therefore, a
long spin-up period is employed for initialising the land surface.

*2.3 Methodologies to detect the nature of diurnal variation*
Harmonic analysis has been widely used to quantify the main characteristics
of the diurnal cycle (e.g., Yang and Slingo, 2001; Diro et al., 2012; Mooney et al.,
2017). One particular advantage of harmonic analysis is the estimation of the
explained variance (%) of a specific frequency and its phase from a time series. This
study follows Mooney et al. (2017) by fitting the following function to the NDJFM-
averaged hourly data,
$$R(t) \cong a_{24} \cos\left(\frac{2\pi(t - \phi_{24})}{24}\right) + a_{12} \cos\left(\frac{2\pi(t - \phi_{12})}{12}\right) \quad (1),$$
where $R(t)$ is the hourly variation of total rainfall and $a_{24}$ ($a_{12}$) and $\phi_{24}$ ($\phi_{12}$) are the
amplitude and phase of the diurnal and semi-diurnal cycle, respectively.
The empirical orthogonal function (EOF) analysis is additionally used to
capture the features of the diurnal cycle around Lake Malawi following previous
studies (e.g., Kikuchi and Wang, 2008; Teo et al., 2011).
The EOF analysis is used to identify the dominant spatio-temporal patterns.
For the diurnal cycle, it is known that the first mode represents a stationary dipole
pattern between coastal land and ocean while the second mode identifies a
propagation pattern from land to sea (the EOF patterns and principle component
scores between the first and second modes are out of phase by approximately $\pi/4$,
e.g., Kikuchi and Wang, 2008; Teo et al., 2011). Employing these statistical
methodologies, we will explore the details of the observed and modeled diurnal cycle
around Lake Malawi in Section 3.2 and 3.3. The EOF analysis is adopted into the
diurnal deviation components with,
$$A'(t) = A(t) - \overline{A} \qquad (2),$$

where $A$ is a variable and $t$ is time (hourly). The overbar and prime denote the daily-
mean and daily-deviated components, respectively.

**3. Results**
In this section, we will show the essential features of diurnal cycle of
precipitation around Lake Malawi using satellite observations and WRF simulations.
Additionally, the results of the idealized WRF simulation will be compared and
contrasted with the control simulation to reveal  the role of Lake Malawi in the local
diurnal cycle of precipitation.

*3.1 Climatology*
Firstly, we take an overview of the climatological diurnal cycle of
precipitation around Lake Malawi using TRMM 3B42 which has good temporal
coverage but relatively coarse resolution (temporarily and spatially). Figure 2
illustrates the 3-hourly precipitation obtained by TRMM-3B42 for NDJFM-mean
climatology. Between 00-03 to 06-09 UTC (02-05 to 08-11 LST), the precipitation
over Lake Malawi is enhanced and the precipitation over the surrounding land area
becomes weaker. At 09-12 UTC, the precipitation is suppressed over the entire area.
Later, from 12-15 LST, precipitation is activated over the land surrounding Lake
Malawi. The land precipitation intensifies widely at 15-18 UTC while rainfall over
Lake Malawi is negligible. From 18-21 to 21-00 UTC, the land precipitation is
gradually reduced and precipitation over Lake Malawi commences. That is, around
Lake Malawi there is a well-organized diurnal variation in precipitation. Interestingly,
the magnitude of land and lake precipitation is almost identical (0.9 mm/h).

*3.2 Case study, 2014/15 NDJFM*

In this subsection, the more detailed nature of the diurnal cycle, which is

indicated in the preceding subsection, is investigated with a finer resolution satellite
product and numerical simulation for a case study of November to March in 2014/15.
Figures 3a-e show monthly-mean rainfall for GPM from November to March. In
November, the daily rainfall around Lake Malawi is low compared to the other
months. There is little rainfall over the Southern part of Lake Malawi but there is
some intense rainfall over the northern part of the lake. Rainfall becomes more
intense in December particularly over the centre of Lake Malawi. Precipitation peaks
in January and it is very intense in the entire domain with rainfall over Lake Malawi
reaching ~ 22 mm/h. From February to March, the precipitation over land decreases
while the lake precipitation over the lake remains strong, especially, in the central
area (around 18 mm/h). The precipitation over Lake Malawi is not distributed
homogeneously, but it seems that there is a dependency on location: the precipitation
is intense in the central part of the lake in December to March, in particular, the
precipitation spreads broadly around the centre of the lake. In the northern and
southern edges of the lake, there are also moderate peaks of the precipitation in
Feruary and March. These distributions might be determined by several factors (for
example, lake surface temperature), which is a highly complex process and beyond
the scope of this study. Fig. 3f-j show that WRF-CTL can capture the seasonal march
of larger-scale precipitation. However, the land precipitation tends to be
overestimated, in particular, from January to February. This overestimation might be
due to the high topography (higher than 2300m) around Lake Malawi (see section 4).
WRF-CTL successfully reproducse the intense lake precipitation from November to
March.

*3.3 Harmonic analysis*

Figure 4 shows the key characteristics of the diurnal cycle of precipitation

obtained by harmonics analysis (see Section 2.3) for NDJFM-mean hourly data of
GPM and WRF-CTL. Over Lake Malawi, the GPM observed sub-daily variations are
dominated by the diurnal cycle as shown in Fig. 4a (about 70-80% of explained
variance). Other dominant diurnal cycles are seen along the coast of Lake Malawi and
to the northeast of Lake Malawi, around 10°S and 35-36°E with a similar explained
variance. In WRF-CTL the dominant diurnal variation are captured well over Lake
Malawi with 60-70 % of explained variance in Fig. 4e. Although the strength of the
diurnal signal over the coastal region tends to be underestimated to some extent, the
terrestrial diurnal cycle is well represented in WRF-CTL in terms of the explained
variance.

The largest amplitudes of the diurnal cycle ($a_{24}$ in Eq. 1) are observed over

Lake Malawi (up to 0.5 mm/h) and its coastal region (Fig. 4b). Over land, the
amplitude is relatively large to the northeast of the Lake (0.2-0.3 mm) where the
diurnal cycle dominates the sub-daily variations (Fig. 4a). This distribution of
amplitude is fairly well simulated by WRF-CTL in Fig. 4f. However, over Lake
Malawi, the amplitude is lower than observed while the amplitude over land to the
Northeast is too large (0.5 mm). This is consistent with the overestimated monthly-
mean precipitation in Fig. 3. The observed phase of the diurnal cycle ($\phi_{24}$ in Eq. 1)
shows a clear contrast over the lake and land in Fig. 4c; the maximum peak of the
precipitation is at 02-03 UTC over the lake and surrounding coastal area and at 13-14
UTC over the land north of the lake where terrestrial precipitation is relatively large
(Figs. 4a and b). This result is consistent with the climatological overview in the
previous subsection. The timing of the WRF-simulated diurnal cycle in Fig. 4g agrees
reasonably with that of the observations. Over the lake, the peak time is slightly late
especially in the south (at 03-05 UTC) compared to the observations and the land
precipitation is maximized at 13-14 UTC to the north of the lake. However, over the
central eastern coastal region, the timing of the rainfall is incorrectly simulated.

In Figs. 4d and h, the explained variance of semi-diurnal cycle is given for

GPM and WRF-CTL. Neither product shows a clear semi-diurnal cycle around Lake
Malawi although there are some spots with relatively high variance of 40-50 %. These
results suggest that the sub-daily variations in rainfall are mainly associated with the
diurnal cycle over and around the lake while semi-diurnal cycle is almost negligible.

Figures 4i-l show the characteristics of the diurnal cycle of precipitation

calculated by the harmonic analysis (Eq.1) for WRF-NOLM. Compared to WRF-CTL
(Fig. 4e), the explained variance of the diurnal cycle is almost identical around Lake
Malawi, in particular, to the northeast of the lake. Over the lake, the variance of the
diurnal cycle is reduced remarkably in the southern part of the lake, which drops
down to 20-30% in Fig. 5i (50-60% in WRF-CTL, shown in Fig. 4e). To the north of
the lake, the diurnal cycle persists despite the absence of the lake. However, the
amplitude of the diurnal cycle shrinks over the entire lake in Fig. 4j. Most notably, the
reduction is largest in the central part and the northern part of the lake (a decrease
from 0.5 mm/h to 0.1-0.2 mm/h) even though the variance of the diurnal cycle is still
comparable to the WRF-CTL case. Over land, the diurnal amplitude is largely
unchanged when Lake Malawi is removed; this is most evident overland to the
northeast of the lake. The phase of the diurnal cycle is also modified over the lake. Its
peak is slightly earlier (around 02-03 UTC) than WRF-CTL (comparison between
Fig. 4g and 4k). In the southern shore of the lake (where the diurnal cycle almost
disappears), the phase is noisy with respect to WRF-CTL. The component of the
semi-diurnal cycle is almost identical with that in WRF-CTL and the semi-diurnal
cycle is not of importance in the sub-daily variations (Fig. 4l).

*3.4 EOF analysis*

The dominant spatio-temporal pattern of variation is provided through the

EOF analysis in Fig. 5. The EOF first mode of GPM shows a clear contrast between
the land and lake (Fig. 5a). The amplitude is larger over the lake than over the land
suggesting that the lake rainfall is more intense than the land rainfall. The coastal land
rainfall synchronizes with the lake rainfall in both eastern and western shores. This
mode explains 53.69 % of the total variance and its principal component (PC) score
(Fig. 5h) shows a distinct diurnal cycle. The peak of rainfall over land is between 12-
17 UTC and that of the lake rainfall over the lake is between 23-03 UTC. This seesaw
pattern of daytime rainfall over land and nighttime rainfall over the lake is quite
similar to the pattern described by sea-land contrast in the tropics (e.g., Teo et al.,
2011; Bhatt et al., 2016). The EOF second mode has 15.77% of the total variance and
its spatial pattern and PC score do not indicate a propagation mode from land to lake
(not shown). The PC score seems a semi-diurnal cycle and the spatial pattern is quite
spotty and it appears to be unrelated to Lake Malawi. Its amplitude is considerably
smaller than that of the first mode.

WRF-CTL represents well the sharp contrasting spatial pattern between the

land and lake in Fig. 5b as an EOF first mode (the explained variance is 41.51%).
However, as shown in Figs. 3 and 4, the amplitude of the land precipitation is
overestimated and coastal terrestrial rainfall synchronizing with the lake rainfall does
not spread widely compared to the observation although there is some coastal land
precipitation occurring simultaneously with the lake precipitation. While the PC score
of first mode is roughly consistent with that of observation (Figs. 5h and i), the phase
is somewhat shifted: the peak of the nighttime rainfall is around 03-07 UTC (later
than the observation) and that of daytime is around 12-14 UTC, which is slightly
earlier than the observation. In particular, the earlier simulated peak in the daytime
precipitation is a common issue in regional climate modeling (e.g., Nikulin et al.
2012; Pohl et al., 2014; Mooney et al., 2016, 2017; Koseki et al., 2018). Similar to the
GPM observations, WRF-CTL does not show any clear propagation mode by the
second mode and the large variation is limited in some small areas (its variance is
18.36 %) although the PC score of the second mode is lagged by approximately $\pi/4$
(not shown).

The modeled surface zonal wind shows an interesting distribution by the EOF

first mode in Fig. 5c: the lake shore is encompassed by the narrow bands of the
negative and positive daily anomalies of surface zonal wind (77.88% of the total
variance) and those bands spread over Lake Malawi. Combined with its PC score it
can be interpreted that the outgoing flow from the Lake is maximized between 09-13
UTC (Fig. 5i) and the incoming flow into the Lake is dominant between 21-03 UTC.
This diurnal-varying circulation is consistent with a well-characterized lake-land
breeze (e.g., Keen and Lyons, 1978; Crosman and Horel, 2010). The PC score of the
surface zonal wind leads that of the precipitation by approximately 3 hours. The
surface meridional wind also shows a remarkable pattern by the EOF first mode (61.
46% of the total variance) in Fig. 5d: with a macroscopic view, there is a dipole mode
of positive in the north and negative in the south of Lake Malawi. Combining it with
the PC score (Fig. 5i), there is an outgoing/incoming flow of meridional surface wind
during daytime/nighttime respectively. The EOF $1^{st}$ mode of meridional wind varies
approximately with the zonal wind as shown in Fig. 5i.

The EOF 1st mode also shows substantial changes in the diurnal cycle in

WRF-NOLM as shown in Figs. 5e-g; the dipole pattern between the lake and
surrounding terrestrial area almost disappears in the EOF 1st mode and the dominant
variability is only over the land in Fig. 5e. The variance is still 35.60% and the
amplitude over the land is almost identical with that of WRF-CTL in Fig. 5b. While
the harmonic analysis estimates the diurnal cycle independently at each grid cell, the
EOF analysis calculates the most explainable variability in all the selected grids and
therefore, the amplitude at one grid would be affected by that at other grids. That is, in
Fig. 5e, the variability at the lake grids are much smaller than those at land grids,
which is consistent with the reduced amplitude of diurnal variation over the lake in
Fig. 4j. The PC score indicates that the EOF 1st mode is a diurnal cycle in Fig. 5j with
some modification in its peak time. Whereas the EOF 1st mode of surface zonal
winds have the two narrow bands along the lake shore in WRF-NOLM (74.53% of
the total variance), their spreads over the lake are largely diminished on both sides of
the lake shore with respect to that in WRF-CTL (Figs. 5c and 5f). The magnitudes of
the WRF-CTL (Fig. 5i) PC scores are similar to those for WRF-NOLM (Fig. 5j) and
the maximum and minimum of the PC scores for both WRF-CTL and WRF-NOLM

occur during the day and the night, respectively. Similarly, the variability in surface meridional wind is also reduced over the lake as shown in Fig. 5g. However, there is still some evidence of a dipole pattern between the northern and southern part of Lake Malawi as shown in WRF-CTL (Fig. 5d). However, the maximum of PC scores for the meridional wind occurs slightly earlier in the WRF-NOLM (Fig. 5j) simulation compared to the WRF-CTL (Figs. 5i).

*3.5 Nighttime and daytime precipitation*

As witnessed by the harmonic and EOF analyses above (Figs. 4-5), Lake Malawi plays a crucial role in the generation and/or amplification of the diurnal cycle of precipitation. At certain times in the day, the lake's role can be clearer than other times (Fig. 6). During 00-03 UTC, the nocturnal precipitation occurs over Lake Malawi in WRF-CTL (Fig. 6a) but this the lake-anchored precipitation is extensively reduced in WRF-NOLM (Fig. 6b). Its influence is remarkable over the entire lake, in particular, over the northern and central parts of the lake (Fig. 6c). This indicates the importance of Lake Malawi for rainfall over the lake (as concluded by Diallo et al., 2018). Conversely, the surrounding area of the lake experiences a modest reduction in precipitation in the presence of the lake during midnight to early morning. During daytime when the precipitation peak is closely tied to the maximum in local solar heating (11-14 UTC), precipitation is more dominant over the surrounding area of the lake than over the lake in WRF-CTL (Fig. 6d). While precipitation over the lake is quite small, there is some increase in the precipitation over the southern part of the lake in WRF-NOLM, (Fig. 6e). In contrast to the nocturnal precipitation, daytime precipitation is amplified over the southern part of the lake although its response is relatively weaker than that in the nighttime (Figs. 6c and f).

Figure 7 presents the surface horizontal wind and its divergence anomalies

from daily-mean at nighttime and daytime, estimated by Eq. (2). In WRF-CTL, the
incoming flow from the shore toward the lake is detected and the strong convergence
forms over the lake shown in Fig. 7a. These lakeward flows are land breeze
circulations and penetrate deeply into the lake as shown by the EOF analysis (Figs. 5c
and d). The intense nocturnal rainfall (as in Fig. 6a) can be attributed to this strong
convergence over the lake. In WRF-NOLM, the land breezes are extensively
weakened and, as a result, the convergence over the lake shrinks considerably (Fig.
7b). The difference shows clearly that the intensification in the land breeze and
convergence is due to Lake Malawi (Fig. 7c). While the daily-residual component of
the surface wind can be seen not only around the lake but also in the region (Figs. 7a
and b), the influence of the lake on the wind seems to be limited around and over the
lake. During daytime, on the other hand, the outgoing flows, thus, lake breezes are
organized well from the lake outward and this flow is highly divergent over the lake
in WRF-CTL (Fig. 7d). This outgoing circulation can also be seen in WRF-NOLM
(Fig. 7e), but its magnitude is considerably reduced and the flow-forming divergence
is also reduced. The difference during daytime is almost a mirror image of that during
nighttime and it shows that Lake Malawi plays an important role in the diurnal
variations of local wind circulations. The lake surface seems to create a heat contrast
favouring the lake-land breeze circulation in night- and daytime: the surface
temperature over the lake is higher in WRF-CTL ($25.7^{\circ}$C) than in WRF-NOLM
($24.8^{\circ}$C) during nighttime and lower in WRF-CTL ($26.8^{\circ}$C) during the daytime than
in WRF-NOLM ($32.8^{\circ}$C) This behaviour in the surface temperature can create
favourable conditions for more convergence (divergence) and consequently, the
precipitation over the lake is enhanced (suppressed) effectively.

### 3.6 Moisture flux convergence

The preceding subsections have shown that Lake Malawi radically drives the diurnal cycle in precipitation, and local circulations. Since the moisture flux, $\mathbf{U}q$, (here, $\mathbf{U}$ is horizontal wind vector and $q$ is specific humidity at surface) due to the lake-land breeze circulations can be highly related to precipitation, we quantify the surface moisture flux and its diurnal variation. Note that 10m and 2m data are used for horizontal wind and specific humidity in this study. The moisture flux can be subdivided into four components like,

$$\mathbf{U}q(t) = \left(\overline{\mathbf{U}} + \mathbf{U}'(t)\right) \cdot \left(\overline{q} + q'(t)\right) = \overline{\mathbf{U}}\overline{q} + \overline{\mathbf{U}}q'(t) + \mathbf{U}'(t)\overline{q} + \mathbf{U}'(t)q'(t),$$

where the overbar and prime denote daily-mean and daily-deviation as Eq. 2. $\mathbf{U}$ is surface wind vector and $q$ is surface specific humidity. The horizontal divergence of moisture flux is calculated as

$$\nabla \cdot \mathbf{U}q(t) = \underbrace{\nabla \cdot \overline{\mathbf{U}}\overline{q}}_{A} + \underbrace{\nabla \cdot \overline{\mathbf{U}}q'(t)}_{B} + \underbrace{\nabla \cdot \mathbf{U}'(t)\overline{q}}_{C} + \underbrace{\nabla \cdot \mathbf{U}'(t)q'(t)}_{D} \quad (3).$$

The term $A$ is the moisture flux divergence/convergence due to daily-mean wind and humidity, which does not have diurnal variation, but its relevance is more to the moisture flux associated with the Indian winter monsoon over this region. The term $B$ reflects the influence associated with the diurnal variation in the heat flux and the background wind. The term $C$ indicates the contribution due to the lake-land breeze and the daily-mean humidity to the moisture flux divergence/convergence. The final term is attributed to the diurnal variations in local breeze and humidity. Since the term $A$ does not contain any temporal change, only the three terms of $B$, $C$, and $D$ are averaged over Lake Malawi and surrounding area as shown in Fig. 8a.

During nighttime, the moisture flux converges over Lake Malawi and diverges

over the surrounding area mainly by the lake-land breeze circulation and background
humidity in WRF-CTL (term C in Figs. 8b and c). The daily-mean (background)
latent heat flux averaged over the lake grids is 155.2572 and 56.9174 W/m$^2$ for WRF-
CTL and WRF-NOLM, respectively and the lake surface is an important water source
of the local precipitation, depending on the wind conditions and other characteristics
(e.g., topography). The intense moisture flux convergence is responsible for the
nocturnal precipitation as shown in Fig. 8c. Other terms in Eq.3 do not substantially
contribute to the moisture flux divergence/convergence. In WRF-NOLM, the diurnal
varying breeze and background humidity also contributes to the moisture flux
convergence/divergence, but its magnitude is much smaller than that in WRF-CTL in
Figs. 8b and c. Consequently, the precipitation over the lake area is reduced without
Lake Malawi. As shown in Fig. 6c, the precipitation surrounding the lake is somewhat
enhanced in WRF-NOLM during nighttime (although the response of the rainfall is
noisy and weak, the consistency with the response of the moisture flux is reasonable).
During daytime, the lake-land breeze and background humidity are still the main
driver of the moisture flux divergence/convergence over the lake and surrounding
area in Figs. 8d and e. Without the lake, the divergence over the lake and convergence
over the lake shore are weakened, which is consistent with the enhanced (reduced)
daytime rainfall surrounding (over) the lake in WRF_CTL in Fig. 6f. The term $C$ is
mainly contributed by the zonal component, $\partial(uq)/\partial x$, which is about 70 to 80 % of
the total divergence/convergence (not shown).

In both cases of nighttime and daytime, the other terms of $B$ and $D$ in Eq.3 do

not contribute to the diurnal changes in moisture flux divergence/convergence. That
is, the land-lake breeze and the enriched background water vapor due to Lake Malawi
mainly drive the diurnal variations in surface moisture flux and consequently, the
precipitation around Lake Malawi.

**4. Discussion**

The previous section has revealed that Lake Malawi plays a vital role to form

the diurnal variations in land-lake breeze systems and correspondingly, the
precipitation. However, the diurnal cycle of surface winds do not completely
disappear in WRF-NOLM, and there is still a signature of the diurnal cycle detected
even in the absence of the lake. We provide a brief discussion on the possible other
factors of the diurnal cycle around Lake Malawi.

While Lake Malawi is an active driver of the diurnal variations in the local

land-lake breeze circulations, the local breeze circulation residually remains without
Lake Malawi as shown in Fig. 7. As previous research (e.g., Tyson 1968a, 1968b and
Koseki et al. 2018) has shown, complex terrain also induces a diurnal cycle in the
mountain-valley breeze circulation whose mechanism is similar to that for sea-land
and lake-land breezes. As shown in Fig. 9a, Lake Malawi is encompassed by the high-
elevated terrain that is up to 2600m in the northeast. The altitude is below 600 m over
all of Lake Malawi. This difference in the elevation forms the large gradients in the
surface as shown in Figs. 9b and c. In particular, the two narrow bands of steep zonal
gradient run along the eastern and western shore sides. These gradients can drive the
down-hill mountain (incoming toward the lake) and uphill valley (outgoing from the
lake shore) breeze circulations during nighttime and daytime respectively as shown in
Fig. 5b, 7c and 7f. In addition to the lake shore, there are some steep gradients to the
northeast (9°S and 34.5°E) and the southwest (14.5°S and 33.5-34.5°E). Around these
high mountains, there are well-organized mountain and valley breeze circulation
during nighttime and daytime as in Figs. 7a, b, d and e. The daytime precipitation is
enhanced around these regions, that is, the valley breeze can activate the cumulus
convection and precipitation due to the topography-lifting effect (e.g., Joseph et al.,
2008). The overestimated precipitation in the WRF simulations might be caused by an
over sensitive response in convection to this valley breeze circulation.

As previously mentioned, the high topography around Lake Malawi can be

another driver of the diurnal cycle around Lake Malawi. However, there could be
some difference in timing between the diurnal cycle induced by the lake and mountain
due to the difference in heat capacity. Therefore, in WRF_NOLM where the mountain
is only a driver, the peak time of precipitation over the lake differs from that in
WRF_CTL. That is, the diurnal cycle around Lake Malawi is a complicated system
influenced by both the lake and the mountain. Similar mechanisms can be expected in
other places where large lakes are surrounded by high mountains (e.g., Lake
Tanganyika in Tanzania). Future work will investigate explicitly the role of high
terrain in diurnal cycle of precipitation.

In our sensitivity experiment, we used only one land cover type and one soil

type in the lake grid cells. This can slightly infleuce our results, as previous research
(e.g., Bonan, 2008) has shown that changing the land cover from forests to open
spaces (e.g., savanna or croplands) impacts precipitation and temperature. These
differences are driven by changes in parameters associated with each land cover type,
such as, albedo, surface roughness, leaf area index, and root depth. In tropical regions,
changes from forest cover to grass decreases precipitation and increases temperature
by changing the partitioning of the net surface radiation between latent and sensible
heat fluxes (Bonan, 2008; Pitman et al., 2011). In particular, Semazzi and Song
(2001) showed that changing the land cover type from forest to savanna grasslands

reduced precipitation over Mozambique. Consequently, changing the lake cover to a tropical forest instead of svanna in our WRF-NOLM simulation would increase the daytime precipitation in WRF-NOLM, potentially, altering the amplitude of the diurnal cycle. However, it is unlikely that changing the land cover type to forest would impact the phase of the diurnal cycle. Based on this, we hypothesise that changing the lake to a forest cover type instead of savanna in WRF-NOLM, would likely result in slightly smaller differences between WRF-CTL and WRF-NOLM with respect to the amplitude of the diurnal cycle of precipitation but it would have no impact on the phase of the diurnal cycle of precipitation. However, further studies on the importance of the land cover change to the diurnal cycle of precipitation would be necessary to test this hypothesis.

Cumulus convection and associated precipitation are also highly sensitive to and modulated by soil moisture whose features are dependent on land use and soil type (e.g., Walker and Rowntree, 1977; Pielke, 2001; Cook et al., 2006). For example, Sugimoto and Takahashi (2017) suggested that the wetter soil moisture tends to inhibit cumulus convection due to a lower sensible heat flux in South Asia during Indian Summer Monsoon period. In our focusing area, the Indian Winter Monsoon prevails and therefore, it can be anticipated that our results of precipitation and cumulus convection will be changed when the different landuse and soil type are employed in the lake grid cells. Additionally, we have tested only the homogenous distribution of landuse and soil type in the lake grid boxes for the sensitivity experiment. The heterogeneous distribution will modify the distribution of precipitation over the lake. Therefore, further sensitivity experiments with difference land use and soil type would be also interesting to investigate the characteristics of the precipitation and land-atmosphere interactions in this region.

**5. Concluding Remarks**

In this study, we have investigated the diurnal variation of precipitation in summer (November to March) around Lake Malawi using the state-of-the-art satellite products and regional climate model. In a climatological view, TRMM-3B42 shows a clear diurnal cycle of precipitation around Lake Malawi: the precipitation over the lake is more enhanced during midnight to early morning while the surrounding land area experiences a daytime peak with identical amplitudes between the two phases. Such clear contrast between daytime rainfall over the land and nighttime rainfall over the lake can be found over Lake Victoria (Thiery et al., 2016), which is the largest great lake in the African Continent.

The spatially and temporally finer resolution satellite data of GPM and a convection permitting WRF simulation gives a more microscopic view of the diurnal varying precipitation in the area. A harmonic analysis reveals that the diurnal cycle of precipitation is largely dominant over Lake Malawi and to the northeast of the lake and their peak times are almost completely out of phase as suggested by TRMM-3B42. The WRF simulation can capture the diurnal variation in precipitation and reproduce realistic amplitudes of the lake rainfall whilst the land rainfall is overestimated. Analysis of the semi-diurnal cycle shows that the semi-diurnal component is a negligibly small contributor to the diurnal variations. The dominant diurnal variation can also be detected by the EOF analysis as a first principal component (the variance is almost half of the total variance). However, the second modes are not propagating pattern like those identified in Kikuchi and Wang (2008) and Teo et al. (2011). The surface winds also have the dominant first mode of EOF as

diurnal cycle. In particular, the lake-land breeze system is well generated along the lake shore.

Without Lake Malawi, those diurnal variations in precipitation and lake-land breeze are diminished substantially around Lake Malawi: a large part of the diurnal variation in precipitation disappears over the lake region. The magnitude of the lake-land breeze reduces its magnitude over the lake. During nighttime, the land breeze does not penetrate deeply into the lake surface and convergence is not formed effectively. During daytime, the outgoing lake breeze also shrinks and the divergence over the lake is weakened considerably. As a result, the daytime rainfall over the surrounding area becomes relatively moderate in the absence of the lake. Basically, Lake Malawi creates a thermal contrast between the lake and land surface and this contrast can drive a local lake-land breeze circulation (e.g., Steyn 2003; Kruit et al., 2004; Crosman and Horel, 2010). As Diallo et al. (2018) suggested, Lake Malawi is a source of water vapour and enhances the precipitation. The combination of lake-land breeze and enriched background water vapour is the main contributor to the diurnal cycle  the surface moisture flux and consequently that in the precipitation.

Besides Lake Malawi, the steep gradient associated with high topographies encompassing Lake Malawi also induces a diurnal cycle in the local circulation of the mountain-valley breezes. Due to this breeze system, the diurnal cycle of the terrestrial rainfall survives with identical amplitude in the presence and absence of Lake Malawi. That is, the diurnal variation around Lake Malawi forms a combination of the two independent systems of lake-land and mountain-valley breezes.

Based on the analysis of satellite observations and numerical simulations, we conclude that Lake Malawi plays a central role in the remarkable diurnal cycle of precipitation and local circulation in summer. Such information is useful for other

fields such as agriculture and hydropower energy to have more efficient water
resources management. For example, Kumambala and Ervine (2010) reviewed the
water resources related to Lake Malawi and Shire River and its sensitivity of future
climate change using water balance models (e.g., Kebede et al., 2006). The diurnal
variations in precipitation can influence the variables of water balance model such as
rainfall, lake level and outflow from the lake directly. Therefore, our new findings in
this study are informative to the community of water balance models for more
accurate estimation of water resources of Lake Malawi.

This study is mainly a case study in only one particular year. Therefore,

longer studies on the interaction of large-scale monsoon circulations with the diurnal
cycle would be highly desirable. Further analysis should be undertaken on the climate
variability of the large-scale monsoon circulation and its impacts on the diurnal cycle
of precipitation, and the associated terrestrial hydrological processes. Thiery et al.
(2016) have shown that the extreme rainfall due to Lake Victoria is modified by
future climate change. Since Lake Victoria and Lake Malawi are located in the same
tropical region, similar influence of lake-induced precipitation can be expected. Such
insights can help mitigate natural disasters of flooding and drought in this region.

**Acknowledgement**
The authors greatly appreciate two reviewers, Dr. Ryan Teuling and Dr. Femke
Jansen at Wageningen University for their quite constructive and useful comments on
the manuscript. The computational resource of this study is supported by Norwegian
High-Performance Computing Program resources (NN9039K, NS9039K, NN9385K,
NS9207k). S. Koseki is supported by the European Union Seventh Framework
Programme (EU-FP7/2007-2013) PREFACE (Grant Agreement No. 603521), ERC
STERCP project (Grant Agreement No. 648982), and from the Research Council of
Norway (233680/E10). P. A. Mooney gratefully acknowledges funding from the
Research Council of Norway (Grant No. 268243).

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

**Figures**

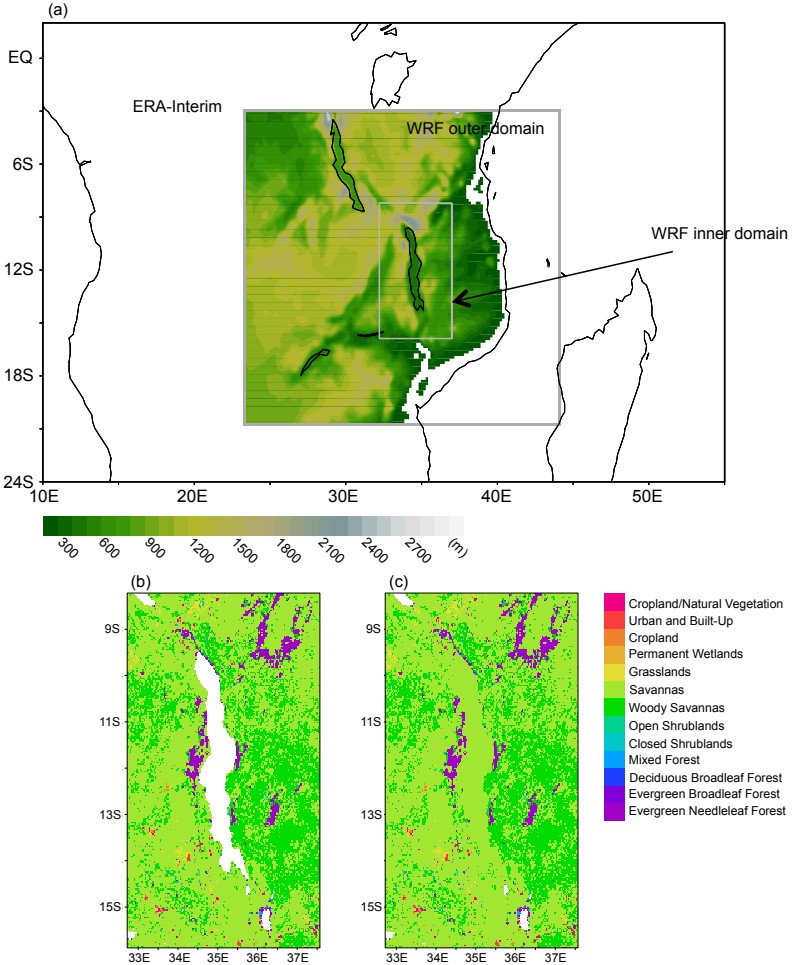

**Figure 1.**
(a) Domains for WRF simulations with terrain height obtained from GTOPO30. (b) and (c) landuse index of the boundary condition for the inner domain of WRF-CTL and WRF-NOLM, respectively


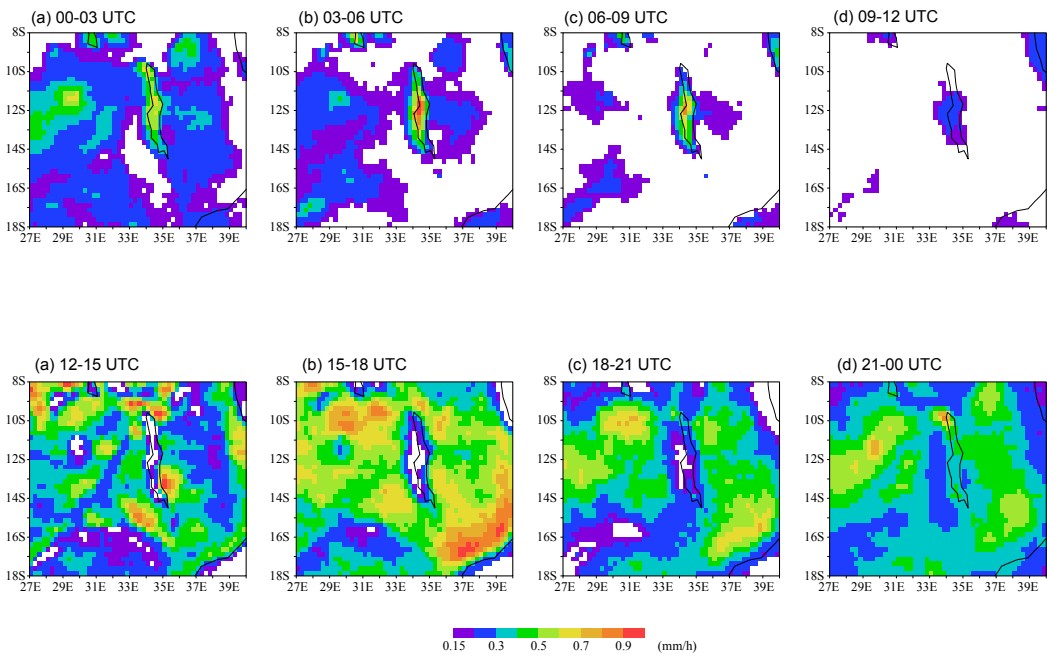

**Figure 2.**
Climatological 3-hourly precipitation of TRMM-3B42 in NDJFM (1998-2012). The white color is precipitation less than 0.15mm/h.


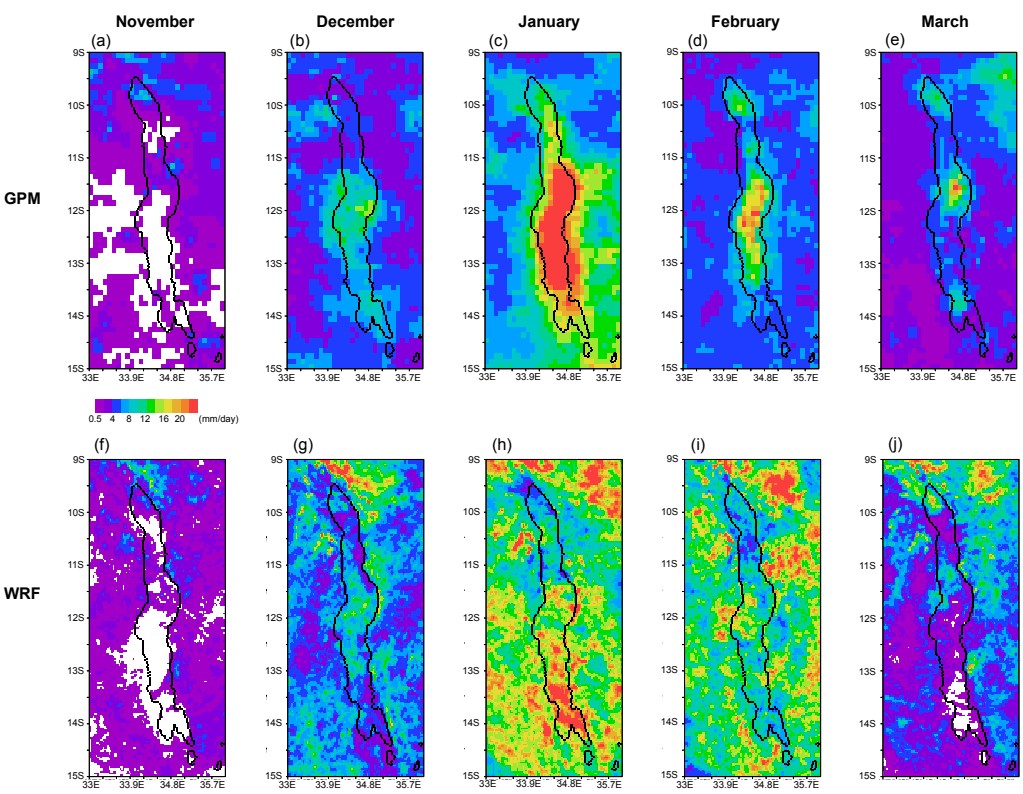

**Figure 3.**
Monthly-mean precipitation of (top) GPM and (bottom) WRF-CTL from November
to March in 2014/15. The white color is precipitation less than 0.5 mm/day.


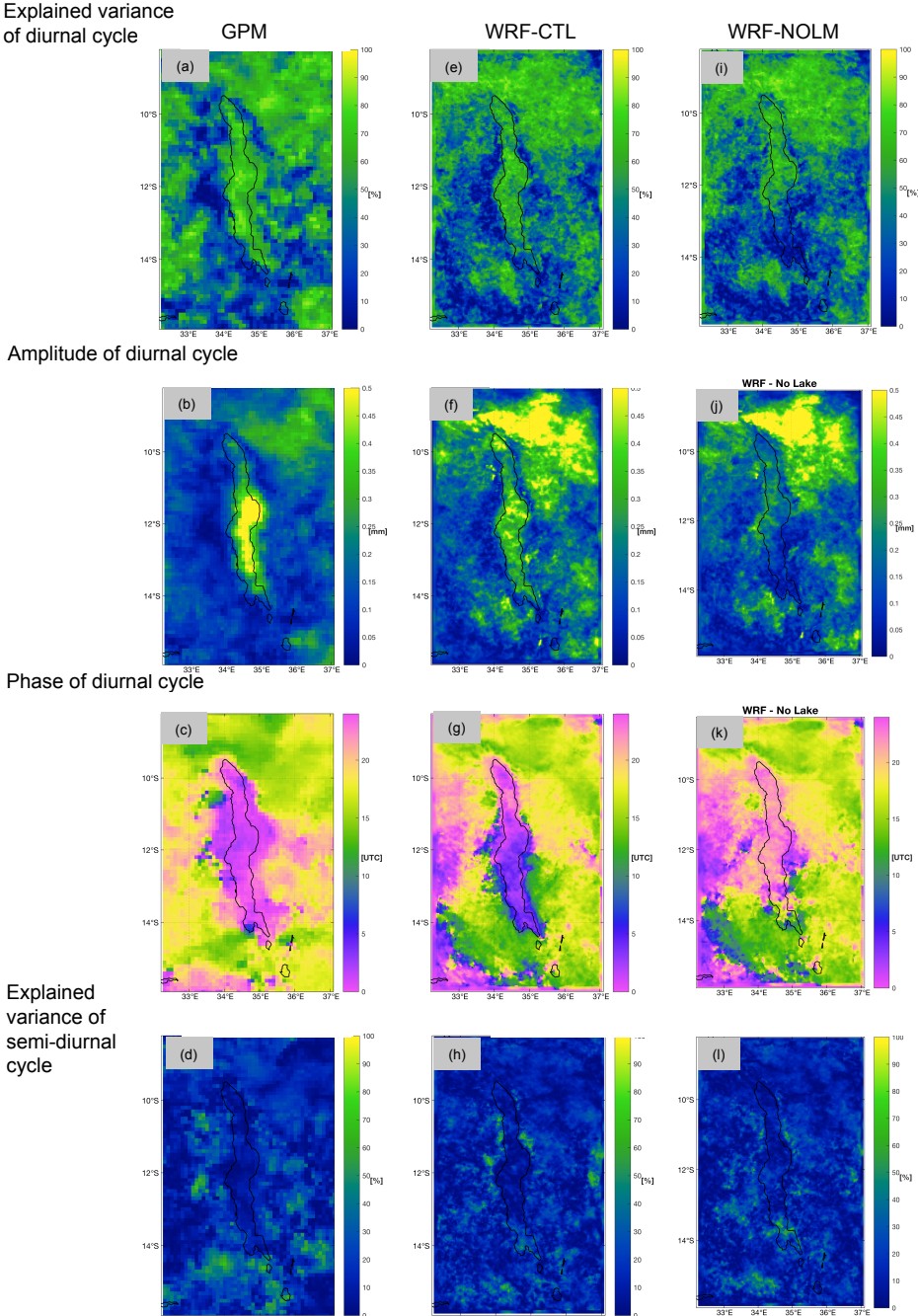

**Figure 4.**
Characteristics of daily-scale temporal variation in precipitation estimated by harmonic analysis for (1st row) explained variance of diurnal cycle, (2nd row) amplitude of diurnal cycle, (3rd row) phase of diurnal cycle, and (4th row) explained variance of semi-diurnal cycle for (left) GPM, (middle) WRF-CTL, and (right) WRF-NOLM, respectively.







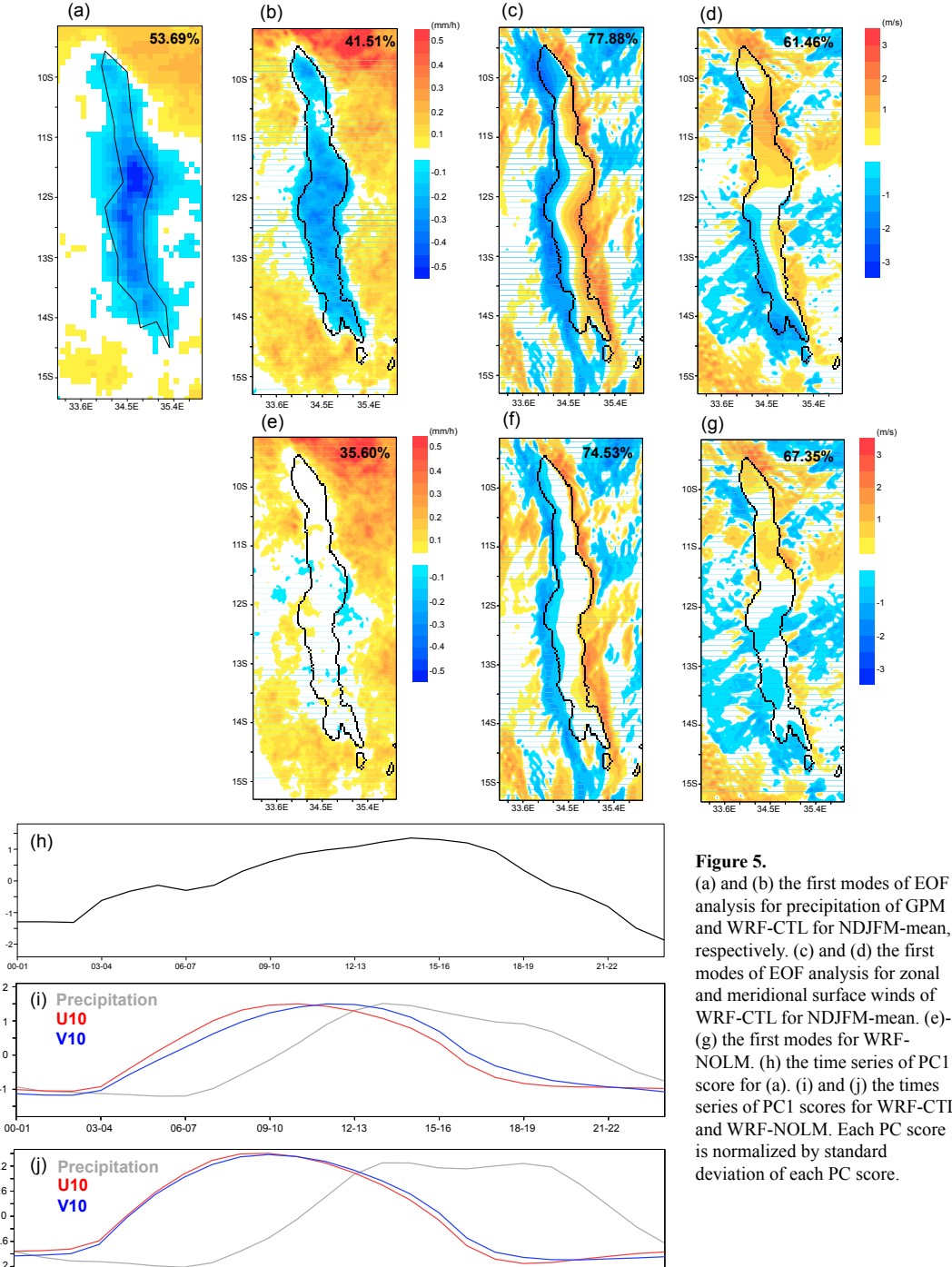

**Figure 5.**
(a) and (b) the first modes of EOF analysis for precipitation of GPM and WRF-CTL for NDJFM-mean, respectively. (c) and (d) the first modes of EOF analysis for zonal and meridional surface winds of WRF-CTL for NDJFM-mean. (e)-(g) the first modes for WRF-NOLM. (h) the time series of PC1 score for (a). (i) and (j) the times series of PC1 scores for WRF-CTL and WRF-NOLM. Each PC score is normalized by standard deviation of each PC score.


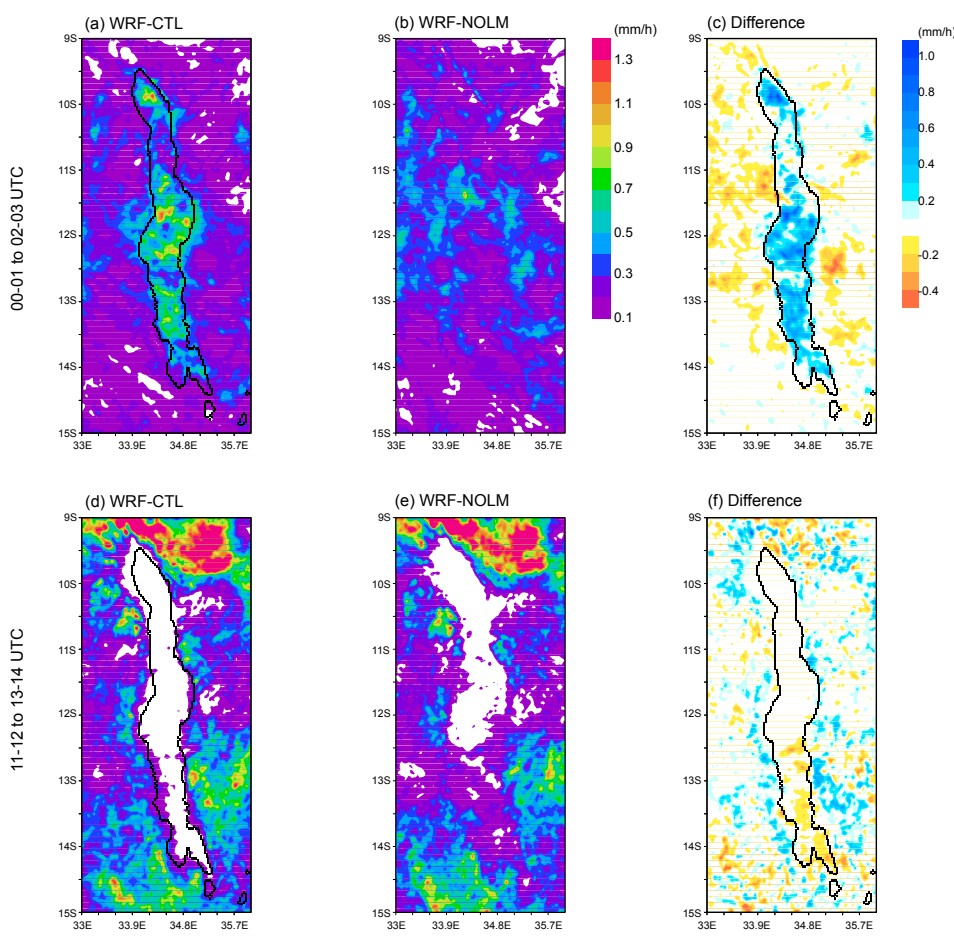

**Figure 6.**
Nighttime mean of precipitation of WRF-CTL and WRF-NOLM in (a) and (b), respectively and its difference (WRF-CTL minus WRF-NOLM) in (c). (d)-(f) same as (a)-(c), but for daytime mean.


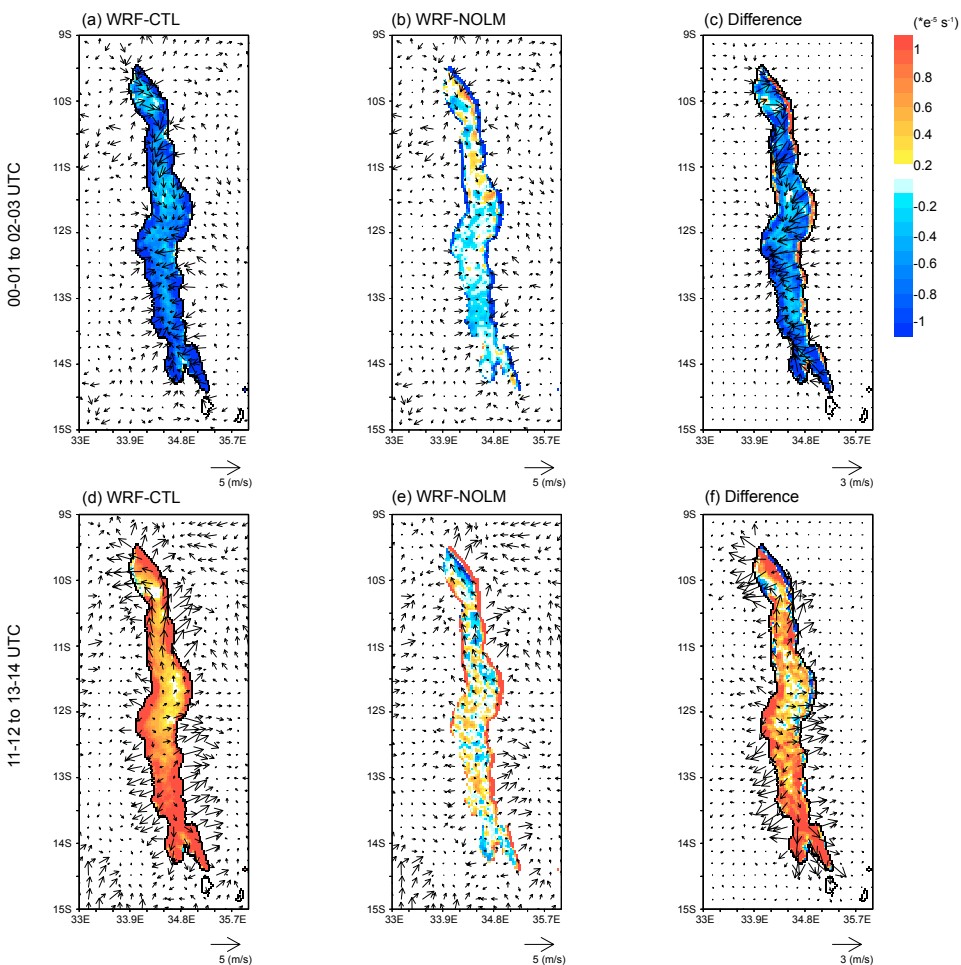

**Figure 7.**
Same as Fig.6, but for surface horizontal winds (arrows) and its divergence (color). Note that the surface winds and its divergence are anomalies from daily-mean values.





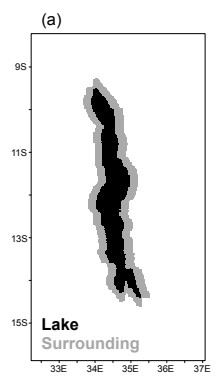

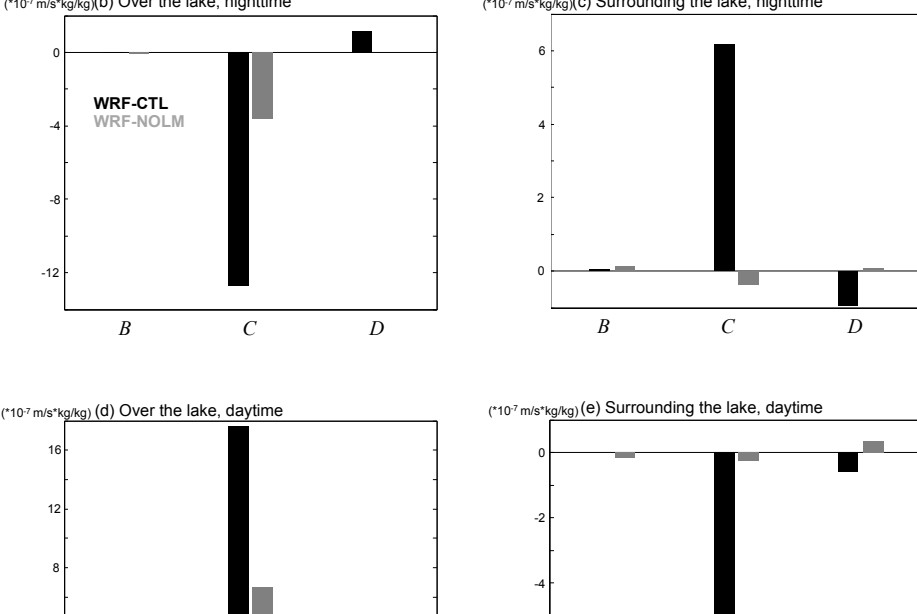

**Figure 8.**
(a) Grids of the lake (black) and surrounding area (gray) for area-averaging. The area-averaged three components of moisture flux divergence in the equation 3 for (b) over the lake, nighttime (00-01 to 02-03 UTC), (c) surrounding the lake, nighttime, (d) over the lake, daytime (11-12 to 13-14 UTC), and (e) surrounding the lake, daytime for WRF-CTL (black) and WRF-NOLM (gray), respectively.


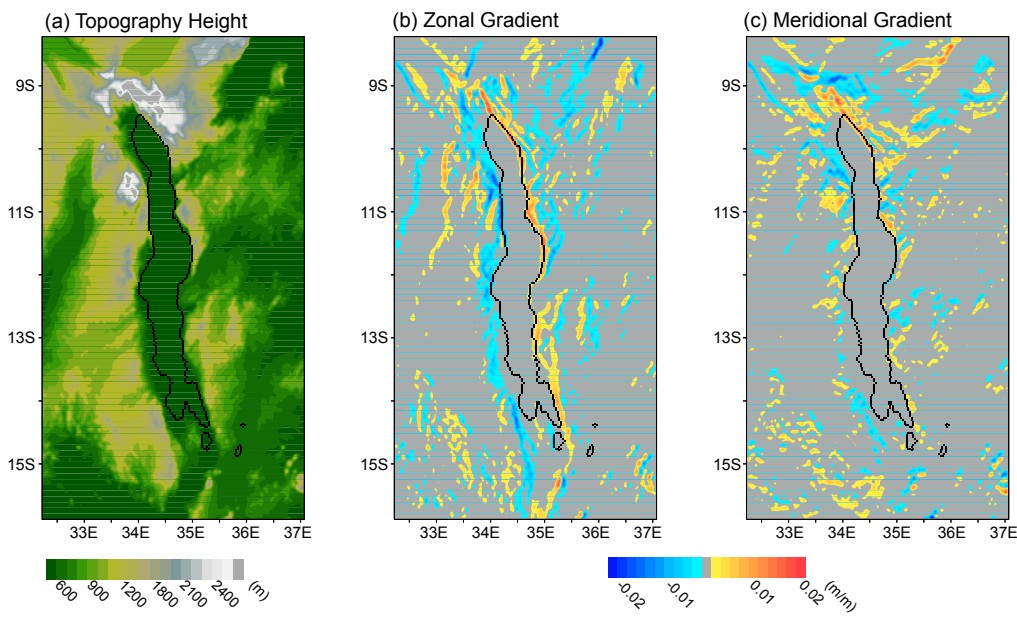

**Figure 9.**
The distribution of topography around Lake Malawi. (a) Topographic altitude in WRF inner domain and its zonal and meridional gradients in (b) and (c).
