# Peer review of "Influences of Lake Malawi"

_Hydrology and Earth System Sciences, 2018_

## Referee Comment (RC1) · Femke Jansen (Referee) · 13 Feb 2019

The manuscript by Koseki and Mooney presents an interesting study on the diurnal variations and patterns of precipitation around Lake Malawi during summer. The authors investigate this cycle and the influence the lake itself has on this cycle, using satellite products and WRF simulations (with and without the lake). To identify and extract the diurnal variations and patterns, and the main contributors to these variations, the authors use harmonic analysis and empirical orthogonal function (EOF) analysis. It was found that the lake has a substantial effect on the diurnal cycle of precipitation due to enhancement of the lake-land breeze circulation. The overall analysis performed in this study is clear, straightforward and concise, and the results give a good overview of the processes responsible for the diurnal variations of precipitation found and the

influence of Lake Malawi. I think this makes the study potentially interesting as it can contribute to our knowledge on the influence of lakes on meteorological variables and consequently on water resources in their direct surroundings. However, I have a few questions and remarks that can improve the clarity and focus of the manuscript before publishing. Therefore, my suggestion is that the manuscript needs in between minor and major revisions, which mainly focus on the structure of the discussion section and on textual improvements.

**General comments**

The first thing that I noticed when opening the manuscript is that the title of the manuscript does not reflect clearly that the focus of the authors is to study the diurnal variations of precipitation. I think it is important to change this and make it more specific to avoid misinterpretation by the readers.

According to the setup of the experiment and the aim stated in the introduction, the focus of this study is quantifying the effect of Lake Malawi on the diurnal cycle of precipitation in the area. However, in the discussion section the focus lies almost completely on the attempt to explain the remaining pattern of precipitation present in case when Lake Malawi is synthetically removed in the model simulation (WRF-NOLM). I would be more interested to first know what the authors learned about the influence of a lake on the regional precipitation patterns based on the analysis they did, and whether the authors think that the same principles apply to other lakes, rather than starting to focus on explaining the remaining pattern (lines 399-403), which lies outside the scope of the aim the authors stated in the introduction (lines 88-91). This means that the discussion section should be revised focussing on the main points that the authors want to state, starting with (1) explaining/quantifying the influence of Lake Malawi on the diurnal variation of precipitation (this can include section 4.1)
which indeed focusses on the effect of the presence of the lake), (2) whether this is generally applicable to other lakes, and then (3) as final remark the authors can write a paragraph on giving examples of factors that may influence the remaining observed patterns in precipitation and local breeze circulations. However, if the authors want to focus on explaining the remaining patterns, which then also should be clearly stated in the aim and introduction, then I would suggest to additional quantify the topographical impact around Lake Malawi (section 4.2) on the local breeze circulation. This can be done by synthetically removing the strong topographical gradients around the lake.

What causes the peak to come slightly earlier in the case without the lake compared to the situation with the lake (lines 280/281)? Can this be expected at other lakes as well? This can be included in the discussion section.

Differences were found between precipitation over the northern part, the central part and the southern part of Lake Malawi. However, the authors do not give an explanation for this. It would be interesting and important to elaborate on this. Is this caused by the characteristics of the lake itself (e.g. bathymetry, mixing), or surrounding topography or land use?

Another general comment that I would like to make is that the English writing is poor in large parts of the current manuscript. This needs to be polished and edited before publishing.

**Specific comments**

At some of the figures the axis labels are missing, i.e. Fig. 5h,i,j, 10. Please be precise in this; describing it in the caption of the figure is not sufficient.
Please include a DEM from the area as background information to the area. This will give the reader a better understanding of the regional landscape.

This comment concerns the use of cumulus parameterization. The authors state that the cumulus parameterization is switched off in the inner domain (lines 139/140). Later they state that in 'this' region the convective scheme Betts-Miller-Janjíc is used (lines 141-144). It does not become clear from this explanation what the authors have used where. Please write this more clearly.

Line 408/409; this argumentation is not precise. The increased humidity of the air through enhanced latent heat fluxes can indeed be a source of precipitation. But depending on wind conditions and other (topographical) characteristics, it not necessarily falls within the studied region.

Line 416; technically it is not the heat capacity of the water surface only, but of the whole water body.

Lines 487-488 493-494; in the first 2 lines I refer to the authors write that the precipitation in the surroundings are enhanced in the situation without the lake. In the latter 2 lines I refer to this seems to be contradicted 'the enhanced and weakened precipitation over the lake and surrounding area'. If this sentence is missing a 'respectively', then it is not consistent with the above 2 lines. Without the word 'respectively' the sentence is not correct as precipitation cannot be enhanced and weakened at the same time.

Technical corrections

L. 33; southeasten -> southeastern

L. 40, 72, 174; quite is not specific enough, now the meaning can be interpreted in
different ways.

L. 44; Reorder the sentence; starting a paragraph with 'In addition' is not common.

- L. 49; creates -> create
- L. 67; brackets around Kumambala and Ervine, 2010
- L. 72; are -> is
- L. 73, 298, 422; On the other hand, this suggests that in the previous sentences on the one hand has been used, which is not the case here.
- L. 74; for -> in
- L. 80; huge is informal language -> large
- L. 84; remove comma
- L. 89; in the regional precipitation -> of regional precipitation
- L. 90; with -> using
- L. 91; and a numerical regional model
- L. 91; Using a satellite product with a relatively coarse spatial resolution, ....
- L. 99; observations
- L. 103; results simulation? -> of the simulation results
- L. 111, 115; superior -> high (and specify the temporal coverage (in case of line 111)).
- L. 116; The temporal resolution of the original Level-3 data is 30 minutes ....
- L. 144; Over the land and lake grids that/which are based on...
- L. 148; in -> at
- L. 160; grids is set to
- L. 165; of the outer domain .... are exactly the same
- L. 167; output of 5 months from November 2014 to March 2015; that is ....
- L. 175; in previous studies (remove the)
- L. 176; The use of harmonic analysis
- L. 181; phase of the diurnal
- L. 183; utilized -> used
- L. 184; remove the between following and previous
- L. 196; where A is a variable and t is time
| L. 209; intense widely? What is meant here?
L. 218; are -> is
L. 218; finer resolution satellite product                                                                                                                                                                                                                                                                                                                                                  | HESSD                    |
|-----------------------------------------------------------------------------------------------------------------------------------------------------------------------------------------------------------------------------------------------------------------------------------------------------------------------------------------------------------------------------------------------------------------------------------------------------------------|--------------------------|
|  <li>L. 221; 'quite modest'; be more specific</li> <li>L. 221; around Lake Malawi, (add comma)</li> <li>L. 222, 223; add part of before Lake Malawi</li> <li>L. 225/226; to which precipitation is the first referred to?</li> <li>L. 231; December should be November?</li> <li>L. 246; remove in and place Fig. 4b between brackets</li> <li>L. 277; down -> a decrease</li> <li>L. 294; obvious -> distinct</li> <li>L. 204; of > in</li>  | Interactive
comment   |
| L. 304, 01 > 15
L. 331; now is says 2x south; the first should be north I think
L. 333/334; what are the outgoing/incoming words referring to? Referring to
daytime/night-time? Then respectively should be added.
L. 337; :-> ;                                                                                                                                                                                                                    |                          |
|  <li>L. 351; specify almost identical</li> <li>L. 352; over the lake as shown</li> <li>L. 353; 'some clue' is informal language</li> <li>L. 354/355; Rewrite last sentence; grammar is not correct</li> <li>L. 359; the diurnal cycle of precipitation</li>                                                                                                                                                                                            |                          |
|  <li>L. 369; 'more dominant; does it refer to daytime vs. hight-time, or WRF-CTL vs. WRF-NOLM?</li> <li>L. 385; remove 'everwhere' -> in the surrounding region</li> <li>L. 409; it -> this</li> <li>L. 413; more -> higher</li> <li>L. 414; less -> lower</li>                                                                                                                                                                            | Printer-friendly version |
| L. 416; surface -> body
L. 444; during night-time and daytime respectively                                                                                                                                                                                                                                                                                                                                                                                   |                          |

L. 463/464; comma should be directly after the equation

L. 480; add term C in between brackets, as you are referring to that.

L. 488; is 'relatively' referring to both noisy and weak? Correct order of words accordingly

- L. 504; the diurnal variation of precipitation
- L. 526/527; The magnitude of the lake-land breeze reduces over the lake.
- L. 530; 'weakened intensively' another word for intensively works better I think
- L. 536; a main contributor -> the main contributor
- L. 547; could be -> is

L. 552/553; the diurnal variations of precipitation can influence rainfall? Seems redundant to mention this as an example.

L. 556-560; Very long last sentence. Better to break it up and keep it short and strong.

---

## Referee Comment (RC2) · Ryan Teuling (Referee) · 26 May 2019

First of all, as editor of this manuscript, I would like to apologize for the delay in the review of your manuscript. I had found reviewers that agreed to provide a review within the given timeframe, but it sometimes happens that these review reports are not being submitted in spite of numerous requests. Because of the large pressure on the pool of potential reviewers, I normally aim for the (minimum) number of required reviewers rather than building in a safety margin from the start. Unfortunately this can in some cases lead to delays in the procedure. For this reason, I have decided to provide the second review myself to avoid further delays.

The manuscript provides an interesting analysis of diurnal variability in precipitation and

convection as induced by one of the largest lakes in Africa, Lake Malawi. By comparing RCM simulations with and without the lake present, it is shown that the lake provides an important control on surface energy exchange variability and regional circulation. The study is well designed, and the manuscript and illustrations are generally of high quality. In particular, I appreciate the combination of satellite data analysis and RCM modelling experiments. The results are supported by the evidence provided, and I only have relatively minor suggestions for improvement. These relate to the title, the structure of the manuscript, the focus of the discussion, literature referenced, and some of the illustrations.

General comments

The title doesn't seem to reflect the focus of the manuscript. The manuscript investigates the impact of the lake, not just the diurnal variability. Furthermore the focus is on precipitation rather than all other aspects of diurnal variability, although other aspects are investigated to explain the mechanisms underlying signals in precipitation. Consider changing.

The structure of the manuscript confused me initially. By focussing first on precipitation (a more indirect effect), and discussing impact on surface fluxes and wind fields (more direct effects) only later, the suggestion is raised that the authors use the model merely as a black-box tool by looking at impacts rather than processes. In my view, the manuscript is easier to understand if the Results section starts with material currently presented in the Discussion.

Related to the previous point, I missed a discussion on some important points. In my view, a discussion should focus on the potential impact of methodological choices on the main conclusions, rather than presenting additional results to interpret other results. So I would expect a discussion on the way the lake is removed in the modelling experiment: what would happen if not the lake water surface but the topography of the lakebed was used, or what if other soil or land use types had been chosen? Since no

second experiment was performed in which the topography was removed, a discussion on how the lake and topography interact would be helpful.

While the authors reference a large body of literature on Lake Malawi and the climate of South-eastern Africa, I miss a general overview of the impact of lakes on surface exchange and regional climate, as well as an introduction to, and comparison with, other studies on impact of lakes (like Lake Victoria, see e.g. Thiery et al., Nature Communications 7, 12786) and other surface heterogeneities like soil moisture in Africa (see e.g. Taylor et al., Nature Geoscience 4, 430–433).

While the figures are generally of excellent quality, I suggest to use a bit more variation in display types where possible. For instance, Figure 8 shows very little variability over the lake and can easily be replaced with a bar plot summarizing the 6 average values over the lake. Some figures also lack a clear title sentence in the caption (like Figs 1, 9, 10).

Specific comments

Line 80-81: Most, if not all, places on Earth have diurnal variability in incoming radiation?

Line 84: remove comma after "Although"

Line 88: This study aims to . . .

Line 180: This is not an equation

---

## Author Comment (AC1) · 31 May 2019

**Author Comments to Reviewer#1 Dr. Femke Jansen**

We are very grateful for the many helpful and constructive comments. Below we respond to each of your comments point-by-point.

General comments

**The first thing that I noticed when opening the manuscript is that the title of the manuscript does not reflect clearly that the focus of the authors is to study the diurnal variations of precipitation. I think it is important to change this and make it more specific to avoid misinterpretation by the readers.**

We agree and we have changed the title to "Influence of Lake Malawi on the local precipitation".

**According to the setup of the experiment and the aim stated in the introduction, the focus of this study is quantifying the effect of Lake Malawi on the diurnal cycle of precipitation in the area. However, in the discussion section the focus lies almost completely on the attempt to explain the remaining pattern of precipitation present in case when Lake Malawi is synthetically removed in the model simulation (WRF- NOLM). I would be more interested to first know what the authors learned about the influence of a lake on the regional precipitation patterns based on the analysis they did, and whether the authors think that the same principles apply to other lakes, rather than starting to focus on explaining the remaining pattern (lines 399-403), which lies outside the scope of the aim the authors stated in the introduction (lines 88-91). This means that the discussion section should be revised focussing on the main points that the authors want to state, starting with (1) explaining/quantifying the influence of Lake Malawi on the diurnal variation of precipitation (this can include section 4.1 which indeed focusses on the effect of the presence of the lake), (2) whether this is generally applicable to other lakes, and then (3) as final remark the authors can write a paragraph on giving examples of factors that may influence the remaining observed patterns in precipitation and local breeze circulations. However, if the authors want to focus on explaining the remaining patterns, which then also should be clearly stated in the aim and introduction, then I would suggest to additional quantify the topographical impact around Lake Malawi (section 4.2) on the local breeze circulation. This can be done by synthetically removing the strong topographical gradients around the lake.**

We agree with the reviewer. In this study, we focus mainly on the effects of Lake Malawi on the diurnal cycle of precipitation and surface wind. Without Lake Malawi (WRF-NOLM), there are still some residuals of diurnal variation and we would think that this residual component should be mentioned for the further analysis in the future. Therefore, as the reviewer suggests, we will reconstruct the Section 4.

**What causes the peak to come slightly earlier in the case without the lake compared to the situation with the lake (lines 280/281)? Can this be expected at other lakes as well? This can be included in the discussion section.**

Thank you very much for raising an interesting point. In according to our experiments of WRF, the diurnal cycle around Lake Malawi is driven mainly by Lake Malawi itself, but there are still some residual components of diurnal cycle. As we have discussed,

the high terrains could be another driver of diurnal cycle in this region. However, these two drivers are independent and the timing of diurnal cycle would be difference because of difference in the heat capacity between land surface (mountain-valley) and water body (lake-land). In WRF_NOLM, only the high terrains remain as a driver and therefore, it can be considered that the timing of diurnal cycle over Lake Malawi is also modified. That is, the diurnal cycle around Lake Malawi is a complex system induced by both of lake and mountain and it can be expected that the similar results are obtained around lakes surrounding steep terrains. As a future work, it will be interesting to investigate explicitly the impacts of the high terrains on the diurnal cycle. We will add these discussions in Section 4.3 (please note that this section is for Topography Impact in the revised manuscript).

**Differences were found between precipitation over the northern part, the central part and the southern part of Lake Malawi. However, the authors do not give an explanation for this. It would be interesting and important to elaborate on this. Is this caused by the characteristics of the lake itself (e.g. bathymetry, mixing), or surrounding topography or land use?**

Thank you very much for raising an interesting point. One of possibilities is lake surface temperature. In general, the tropical precipitation is highly-correlated with underlying sea surface temperature (e.g., Graham and Barnett, 1993; Waliser et al., 1993; Sabin et al., 2013; Roxy, 2014; Koseki and Bhatt, 2018). The precipitation over the lake could also follow a similar relationship. However, there are no literatures about it as long as we know. And, as we discussed in the manuscript, the precipitation can be induced by diurnal cycle due to mountains. This indicates that the distribution of precipitation over the lake can be affected by the topography and its driving diurnal cycle. Also, as the reviewer mentions, the bathymetry and mixing in the lake would be also responsible for the distribution of precipitation over the lake. We totally agree with that this point is very interesting and important to understand the distribution of precipitation, however, it is also very difficult to quantify the contribution due to each (possible) factor since it is so complicated and this point is out of scope of this study. Therefore, we will add some desciption on the distribution of precipitation in the revised manuscript.

Specific Comments
**At some of the figures the axis labels are missing, i.e. Fig. 5h,i,j, 10. Please be precise in this; describing it in the caption of the figure is not sufficient.**

We added a unit for Fig.10 (Fig.9 in the revised manuscript). For Figs.5h-j, PC score of EOF modes here is non-dimensional since the score is normalized by the standard deviation of each PC score. We will add this in the caption in Fig.5.

**Please include a DEM from the area as background information to the area. This will give the reader a better understanding of the regional landscape**.

We will add the topography height of GTOPO in Fig.1a.

**This comment concerns the use of cumulus parameterization. The authors state that the cumulus parameterization is switched off in the inner domain (lines 139/140). Later they state that in 'this' region the convective scheme Betts-Miller-Janjíc is used (lines 141-144). It does not become clear from this explanation what the authors have used where. Please write this more clearly.**

We will re-write the text to make it more clear that cumulus schemes are used only in the outer domain.

**Line 408/409; this argumentation is not precise. The increased humidity of the air through enhanced latent heat fluxes can indeed be a source of precipitation. But depending on wind conditions and other (topographical) characteristics, it not necessarily falls within the studied region.**

We agree and we will write the text to reflect this fact.

**Line 416; technically it is not the heat capacity of the water surface only, but of the whole water body.**

We agree and we will write the text to clarify this.

**Lines 487-488 493-494; in the first 2 lines I refer to the authors write that the precipitation in the surroundings are enhanced in the situation without the lake. In the latter 2 lines I refer to this seems to be contradicted 'the enhanced and weakened precipitation over the lake and surrounding area'. If this sentence is missing a 'respectively', then it is not consistent with the above 2 lines. Without the word 'respectively' the sentence is not correct as precipitation cannot be enhanced and weakened at the same time.**

We will re-write this text to clarify: ' … weakened precipitation over the lake and the enhanced precipitaiton in the surrounding area …'.

Technical corrections
All technical corrections have been implemented as suggested.

---

## Author Comment (AC2) · 31 May 2019

**Reply to Reviewer#2 Dr. Ryan Teuling**

**First of all, as editor of this manuscript, I would like to apologize for the delay in the review of your manuscript. I had found reviewers that agreed to provide a review within the given timeframe, but it sometimes happens that these review reports are not being submitted in spite of numerous requests. Because of the large pressure on the pool of potential reviewers, I normally aim for the (minimum) number of required reviewers rather than building in a safety margin from the start. Unfortunately this can in some cases lead to delays in the procedure. For this reason, I have decided to provide the second review myself to avoid further delays.**

We greatly appreciate for the editor's efforts to obtain reviews for on our manuscript under such unfortunate circumstances.

**The manuscript provides an interesting analysis of diurnal variability in precipitation and convection as induced by one of the largest lakes in Africa, Lake Malawi. By comparing RCM simulations with and without the lake present, it is shown that the lake provides an important control on surface energy exchange variability and regional circulation. The study is well designed, and the manuscript and illustrations are generally of high quality. In particular, I appreciate the combination of satellite data analysis and RCM modelling experiments. The results are supported by the evidence provided, and I only have relatively minor suggestions for improvement. These relate to the title, the structure of the manuscript, the focus of the discussion, literature referenced, and some of the illustrations.**

We are very grateful for the many helpful and constructive comments. Below we respond to each comment point-by-point.

General comments

**The title doesn't seem to reflect the focus of the manuscript. The manuscript investigates the impact of the lake, not just the diurnal variability. Furthermore the focus is on precipitation rather than all other aspects of diurnal variability, although other aspects are investigated to explain the mechanisms underlying signals in precipitation. Consider changing.**

We agree with the comment. The reviewer#1 also suggested rethinking of the title of this manuscript. Following both comments, we changed the title of this work to "Influences of Lake Malawi on the local diurnal cycle of precipitation".

**The structure of the manuscript confused me initially. By focussing first on precipitation (a more indirect effect), and discussing impact on surface fluxes and wind fields (more direct effects) only later, the suggestion is raised that the authors use the model merely as a black-box tool by looking at impacts rather than processes. In my view, the manuscript is easier to understand if the Results section starts with material currently presented in the Discussion.**

We agree. Thank you so much for the constructive comment. We re-read the manuscript and we will consider the structure of the manuscript as suggested.

**Related to the previous point, I missed a discussion on some important points. In my view, a discussion should focus on the potential impact of methodological choices on the main conclusions, rather than presenting additional results to interpret other results. So I would expect a discussion on the way the lake is removed in the modelling experiment: what would happen if not the lake water surface but the topography of the lakebed was used, or what if other soil or land use types had been chosen? Since no second experiment was performed in which the topography was removed, a discussion on how the lake and topography interact would be helpful.**

Thank you very much for the useful comments for the discussions. The points suggested by the reviewer are quite important for the local precipitation and diurnal cycle. Regarding topography, we will expand the discussion on the possible impacts of the topography on the diurnal cycle in Section 4.3 (Topography Impact). For the soil type, we will add more discussions on it by referring previous literatures.

**While the authors reference a large body of literature on Lake Malawi and the climate of South-eastern Africa, I miss a general overview of the impact of lakes on surface exchange and regional climate, as well as an introduction to, and comparison with, other studies on impact of lakes (like Lake Victoria, see e.g. Thiery et al., Nature Communications 7, 12786) and other surface heterogeneities like soil moisture in Africa (see e.g. Taylor et al., Nature Geoscience 4, 430–433).**

Thank you very much for the useful comments for the literatures. We will add some overview of the lake's impacts in the introduction and some comparison with other lake cases in the conclusion.

**While the figures are generally of excellent quality, I suggest to use a bit more variation in display types where possible. For instance, Figure 8 shows very little variability over the lake and can easily be replaced with a bar plot summarizing the 6 average values over the lake. Some figures also lack a clear title sentence in the caption (like Figs 1, 9, 10).**

Thank you very much for the suggestions. We will remake Figure 6 with bar plot and improve the captions of those figures.

**Specific comments**

**Line 80-81: Most, if not all, places on Earth have diurnal variability in incoming radia- tion?**

Exactly. We will add "strong" in the sentence to emphasize the radiation in the tropics.

**Line 84: remove comma after "Although"**

We will remove it.

**Line 88: This study aims to . . .**

We will correct it.

**Line 180: This is not an equation**

We will modify the function like, $R(t) = a_{24}\cos(\ldots)$, where $R(t)$ is the hourly variation in the total rainfall.

---

## Author Comment (AC3) · 31 May 2019

In the first general comments of the reviewer, we will change the title of our manuscript to "Influence of Lake Malawi on the local precipitation".

The previous response was wrong and not identical with the respond to the reviewer#1. I apologize for this mistake.
* * *

---

## Author Response (AR1)

**Reply to Reviewer#1 Dr. Femke Jansen**

We are very grateful for the many helpful and constructive comments. Below we respond to each of your comments point-by-point. We have written our responses in blue for the purpose of clarity.

General comments

**The first thing that I noticed when opening the manuscript is that the title of the manuscript does not reflect clearly that the focus of the authors is to study the diurnal variations of precipitation. I think it is important to change this and make it more specific to avoid misinterpretation by the readers.**

We agree and we have changed the title to "Influence of Lake Malawi on the spatial and diurnal variability of local precipitation".

**According to the setup of the experiment and the aim stated in the introduction, the focus of this study is quantifying the effect of Lake Malawi on the diurnal cycle of precipitation in the area. However, in the discussion section the focus lies almost completely on the attempt to explain the remaining pattern of precipitation present in case when Lake Malawi is synthetically removed in the model simulation (WRF- NOLM). I would be more interested to first know what the authors learned about the influence of a lake on the regional precipitation patterns based on the analysis they did, and whether the authors think that the same principles apply to other lakes, rather than starting to focus on explaining the remaining pattern (lines 399-403), which lies outside the scope of the aim the authors stated in the introduction (lines 88-91). This means that the discussion section should be revised focussing on the main points that the authors want to state, starting with (1) explaining/quantifying the influence of Lake Malawi on the diurnal variation of precipitation (this can include section 4.1 which indeed focusses on the effect of the presence of the lake), (2) whether this is generally applicable to other lakes, and then (3) as final remark the authors can write a paragraph on giving examples of factors that may influence the remaining observed patterns in precipitation and local breeze circulations. However, if the authors want to focus on explaining the remaining patterns, which then also should be clearly stated in the aim and introduction, then I would suggest to additional quantify the topographical impact around Lake Malawi (section 4.2) on the local breeze circulation. This can be done by synthetically removing the strong topographical gradients around the lake.**

We agree with the reviewer. In this study, we focus mainly on the effects of Lake Malawi on the diurnal cycle of precipitation and surface wind. Without Lake Malawi (WRF-NOLM), there are still some residuals of diurnal variation and we would think that this residual component should be mentioned for the further analysis in the future. Therefore, as the reviewer suggests, we reconstructed the Section 4. Please see lines 480-509.

**What causes the peak to come slightly earlier in the case without the lake compared to the situation with the lake (lines 280/281)? Can this be expected at other lakes as well? This can be included in the discussion section.**

Thank you very much for raising an interesting point. According to our experiments with WRF, the diurnal cycle around Lake Malawi is driven mainly by Lake Malawi itself, but there are still some residual components which are likely due to the high terrains surrounding the lake. However, these two drivers are independent and the timing of diurnal cycle would change because of the difference in the heat capacity between land surface (mountain-valley) and water body (lake-land). In WRF_NOLM, only the high terrain remains as a driver and therefore, it can be considered that the timing of diurnal cycle over Lake Malawi is also modified. That is, the diurnal cycle around Lake Malawi is induced by both the lake and mountain, and it can be expected that the similar results are obtained in regions where lakes are surrounded steep terrains. It is recommended that future work focuses on the impacts of the high terrains on the diurnal cycle of precipitation. We added this discussion in Section 4 (please note that this section of discussion has been re-constructed following the reviewer#2 comments). Please see lines 500-509.

**Differences were found between precipitation over the northern part, the central part and the southern part of Lake Malawi. However, the authors do not give an explanation for this. It would be interesting and important to elaborate on this. Is this caused by the characteristics of the lake itself (e.g. bathymetry, mixing), or surrounding topography or land use?**

Thank you very much for raising this interesting point. One of possibility is the lake surface temperature. In general, the tropical precipitation is highly-correlated with underlying sea surface temperature (e.g., Graham and Barnett, 1993; Waliser et al., 1993; Sabin et al., 2013; Roxy, 2014; Koseki and Bhatt, 2018). The precipitation over the tropical lakes could also follow a similar relationship. However, to the bets of our knowledge, there is no evidence in the literature to support this hypothesis. As we discussed in the manuscript, the diurnal cycle of precipitation is influenced by the steep mountains. This indicates that the distribution of precipitation over the lake can also be affected by the topography and its associated diurnal cycle. Also, as the reviewer mentions, the bathymetry and mixing in the lake would be also responsible for the distribution of precipitation over the lake. We totally agree that this point is very interesting and important to understand the distribution of precipitation, however, quantifying the exact contribution due to each (possible) factor is beyond the scope of this study. Therefore, we added some desciption on the distribution of precipitation in the revised manuscript. Please see lines 241-248.

Specific Comments
**At some of the figures the axis labels are missing, i.e. Fig. 5h,i,j, 10. Please be precise in this; describing it in the caption of the figure is not sufficient.**

We added a unit for Fig.10 (Fig.8 in the revised manuscript). For Figs.5h-j, PC score of EOF modes here is non-dimensional since the score is normalized by the standard deviation of each PC score. We added this in the caption in Fig.5.

**Please include a DEM from the area as background information to the area. This will give the reader a better understanding of the regional landscape.**

We added the topography height of GTOPO in Fig.1a.

**This comment concerns the use of cumulus parameterization. The authors state that the cumulus parameterization is switched off in the inner domain (lines 139/140). Later they state that in 'this' region the convective scheme Betts-Miller-Janjíc is used (lines 141-144). It does not become clear from this explanation what the authors have used where. Please write this more clearly.**

We have re-written the text to make it more clear that cumulus schemes are used only in the outer domain. Please see lines 147-152.

**Line 408/409; this argumentation is not precise. The increased humidity of the air through enhanced latent heat fluxes can indeed be a source of precipitation. But depending on wind conditions and other (topographical) characteristics, it not necessarily falls within the studied region.**

We agree and we have re-written the text to reflect this fact. But, please note that the structure of this section has been reconstructed to large extent following the reviewer#2 comments. We added some new argument about this in the revised manuscript. Please see lines 447-451.

**Line 416; technically it is not the heat capacity of the water surface only, but of the whole water body.**

We agree. But due to the reconstruction of the discussion section, this sentence has been removed from the revised manuscript.

**Lines 487-488 493-494; in the first 2 lines I refer to the authors write that the precipitation in the surroundings are enhanced in the situation without the lake. In the latter 2 lines I refer to this seems to be contradicted 'the enhanced and weakened precipitation over the lake and surrounding area'. If this sentence is missing a 'respectively', then it is not consistent with the above 2 lines. Without the word 'respectively' the sentence is not correct as precipitation cannot be enhanced and weakened at the same time.**

We have re-written this text to clarify: ' … weakened precipitation over the lake and the enhanced precipitaiton in the surrounding area …'. Please see lines 462-464.

Technical corrections
All technical corrections have been implemented as suggested.

**Reply to Reviewer#2 Dr. Ryan Teuling**

**First of all, as editor of this manuscript, I would like to apologize for the delay in the review of your manuscript. I had found reviewers that agreed to provide a review within the given timeframe, but it sometimes happens that these review reports are not being submitted in spite of numerous requests. Because of the large pressure on the pool of potential reviewers, I normally aim for the (minimum) number of required reviewers rather than building in a safety margin from the start. Unfortunately this can in some cases lead to delays in the procedure. For this reason, I have decided to provide the second review myself to avoid further delays.**

We greatly appreciate for the editor's efforts on our manuscript. We are very happy that the editor addressed our manuscript.

**The manuscript provides an interesting analysis of diurnal variability in precipitation and convection as induced by one of the largest lakes in Africa, Lake Malawi. By comparing RCM simulations with and without the lake present, it is shown that the lake provides an important control on surface energy exchange variability and regional circulation. The study is well designed, and the manuscript and illustrations are generally of high quality. In particular, I appreciate the combination of satellite data analysis and RCM modelling experiments. The results are supported by the evidence provided, and I only have relatively minor suggestions for improvement. These relate to the title, the structure of the manuscript, the focus of the discussion, literature referenced, and some of the illustrations.**

We are very grateful for the many helpful and constructive comments. Below we respond to each of your comments point-by-point. We have written our responses in blue for the purpose of clarity.

General comments

**The title doesn't seem to reflect the focus of the manuscript. The manuscript investigates the impact of the lake, not just the diurnal variability. Furthermore the focus is on precipitation rather than all other aspects of diurnal variability, although other aspects are investigated to explain the mechanisms underlying signals in precipitation. Consider changing.**

We agree with the comment. The reviewer#1 also suggested rethinking of the title of this manuscript. Following both comments, we changed the title of this work to "Influences of Lake Malawi on the spatial and diurnal variability of local precipitation".

**The structure of the manuscript confused me initially. By focussing first on precipitation (a more indirect effect), and discussing impact on surface fluxes and wind fields (more direct effects) only later, the suggestion is raised that the authors use the model merely as a black-box tool by looking at impacts rather than processes. In my view, the manuscript is easier to understand if the Results section starts with material currently presented in the Discussion.**

Thank you so much for the constructive comment. We re-read carefully the manuscript and we reconstructed the sections of results and discussion as follows:

Section of 4.2 "Moisture flux" has been moved to a new section 3.6. Because the moisture flux analysis is explainable for what process influences the lake-induced diurnal precipitation and therefore, it is better to be placed in the results section. Please see lines 423-471.

The section 4.1 "surface heat flux" is reconstructed substantially. According to the moisture flux analysis, the production of daily-anomaly wind and daily-mean humidity is a main source of the diurnal-varying moisture flux. Therefore, we added the value of daily-mean latent heat flux averaged over Lake Malawi in section 3.6. Please see lines 447-451. Other part of section 4.1 has been merged into the section 3.5 because the sensible heat flux and surface temperature can be explainable for the surface circulation difference of Fig.7. We decided to show the lake-averaged value instead of the figure. Please see lines 415-421.

**Related to the previous point, I missed a discussion on some important points. In my view, a discussion should focus on the potential impact of methodological choices on the main conclusions, rather than presenting additional results to interpret other results. So I would expect a discussion on the way the lake is removed in the modelling experiment: what would happen if not the lake water surface but the topography of the lakebed was used, or what if other soil or land use types had been chosen? Since no second experiment was performed in which the topography was removed, a discussion on how the lake and topography interact would be helpful.**

Thank you very much for the useful comments for the discussions. The points suggested by the reviewer are quite important for the local precipitation and diurnal cycle. Regarding topography, we have extended the discussion on the possible impacts of the topography on the diurnal cycle in Section 4. Please see lines 500-509. For the soil type and land use, we have added more discussions on it by referring to previous literatures. Please see lines 510-544.

**While the authors reference a large body of literature on Lake Malawi and the climate of South-eastern Africa, I miss a general overview of the impact of lakes on surface exchange and regional climate, as well as an introduction to, and comparison with, other studies on impact of lakes (like Lake Victoria, see e.g. Thiery et al., Nature Communications 7, 12786) and other surface heterogeneities like soil moisture in Africa (see e.g. Taylor et al., Nature Geoscience 4, 430–433).**

Thank you very much for the useful comments for the literatures. We added some overview of the lake's impacts in the introduction and some comparison with other lake cases in the conclusion. Please see lines 79-86, 553-555, and 606-610.

**While the figures are generally of excellent quality, I suggest to use a bit more variation in display types where possible. For instance, Figure 8 shows very little variability over the lake and can easily be replaced with a bar plot summarizing the 6 average values over the lake. Some figures also lack a clear title sentence in the caption (like Figs 1, 9, 10).**

Thank you very much for the suggestions. As we reply above, Section 4.1 has been reconstructed substantially and the lake-averaged value is shown in the revised manuscript instead of Fig.8. Also, we have added more information in the captions.

**Specific comments**

**Line 80-81: Most, if not all, places on Earth have diurnal variability in incoming radia- tion?**

Exactly. We added "strong" in the sentence to emphasize the radiation in the tropics. Please see line 88.

**Line 84: remove comma after "Although"**

We removed it.

**Line 88: This study aims to . . .**

We corrected it.

**Line 180: This is not an equation**

We will modify the function like, $R(t) = a_{24}\cos(\ldots)$, where $R(t)$ is the hourly variation in the total rainfall. Please see lines 188-189.

---

## Author Response (AR2)

Dear Dr. Ryan Teuling, the editor of Hydrology and Earth System Sciences

We greatly appreciate the editor for kindly addressing our manuscript and we are very happy that our manuscript has been accepted in Hydrology and Earth System Sciences. Now, we upload the final version of manuscript with figures and supplement. Please note that in the uploaded version we added two sections of "Data Availability" and "Author Contributions" before Acknowledgement.

Again, thank you so much for all the processes the editor made his efforts on and the final decision on our manuscript.

Sincerely,
Shunya Koseki

Shunya.Koseki@gfi.uib.no
+47 55 58 98 24
Geophysical Institute,
University of Bergen,
Allegaten 70, Bergen, Norway